# Deterioration of modern concrete structures and asphalt pavements by respiratory action and trace quantities of organic matter

Akihiro Moriyoshi[1]*, Eiji Shibata[2], Masahito Natsuhara[3‡], Kiyoshi Sakai[4‡], Takashi Kondo[5‡], Akihiko Kasahara[6‡]

1 Emeritus Professor Hokkaido University, Sapporo, Japan, 2 Department of Health and Psychosocial Medicine, School of Medicine, Aichi Medical University, Nagakute, Aichi, Japan, 3 Shimadzu Corporation, Nishinokyo Kuwabara-cho, Nakagyo-ku, Kyoto, Japan, 4 Graduate School of Medical Sciences, Kawasumi, Mizuho-cho, Nagoya City University, Mizuho-ku, Nagoya, Japan, 5 National Institute of Technology, Tomakomai College, Tomakomai, Hokkaido, Japan, 6 Green Consultant Co. Ltd., Eniwa, Japan

☺ These authors contributed equally to this work.
‡ These authors also contributed equally to this work
* moriyosi@eng.hokudai.ac.jp

**Data Availability Statement:** The authors have uploaded and registered their data to the repository of Hokkaido University (HUSCAP) https://www.

## Abstract

In concrete structures (concrete), damage from cracks, deterioration, amorphization, and delamination occur in some structures, causing disaggregation (concrete changed to very fine particles) and hollowing out of the concrete. In concrete pavements, damage from large amounts of pop-out of aggregate occurs from the surface of the concrete pavement 4–5 hours after spraying of snow melting agent on the surface of the pavement. The damage from disaggregation, blistering, cracks, and peeling-off of a surface course have also been observed in asphalt runways and highways. The damage from disaggregation, cracks and pop-out of aggregate in asphalt pavements and concrete structures have long been seen as strange and unexpected and have defied explanation. As a result of examinations in various experiments, it was concluded that all of the unexplained kinds of damage of both asphalt pavements and concrete structures were caused by Trace Quantities of Organic Matter (TQOM), Air Entrained (AE) water reducing agent in air and/or cement, and surfactant in snow melting agent. The emission sources of TQOM and these organic substances were also identified by chemical analysis for these unexpected and unexplained phenomena. The TQOM includes phthalate compounds (phthalates in the following), amine compounds, phosphate compounds, snow melting agent and Sodium Polyoxyethylene Nonyl phenyl Ether Sulfate (SPNES). SPNES is a surfactant in windshield washer fluid for automobiles. We found that the water content and content of organic matter in damaged asphalt pavements and concrete structures are also important indicators for the damage. Further, a new evaluation method for amorphization was proposed in this study and it appears suitable for evaluating the safety of concrete structures along roads which were exposed to TQOM in severely air-polluted environments.

hokudai.ac.jp/research/research/huscap/. Please see the following entries included in the paper: http://hdl.handle.net/2115/39839, http://hdl.handle.net/2115/39838, http://hdl.handle.net/2115/39836, http://hdl.handle.net/2115/43994, http://hdl.handle.net/2115/42824, http://hdl.handle.net/2115/17149, http://hdl.handle.net/2115/39837, http://hdl.handle.net/2115/40067.

**Funding:** Shimadzu Corporation provided support for this study in the form of salary for Mr. Masahito Natsuhara (MN), and Green Consultant Co. Ltd. provided support for this study in the form of salary for Dr. Akihiko Kasahara (AK). The specific roles of these authors are articulated in the 'author contributions' section. The funders had no role in study design, data collection and analysis, decision to publish, or preparation of the manuscript.

**Competing interests:** The authors have read the journal's policy and the authors of this manuscript have the following competing interests: Author NM is the technician of CT in Shimadzu Corporation. He contributed the measurement using micro focus CT scanner for specimen of concrete. Author AK had many empirical works in the field works in NIPPO Corporation for asphalt pavements and concrete structures for long years. He contributed the technical discussions for concrete structures. NM and AK have declared that this project does not violate the conflict of interest. MN is a paid employee of Shimadzu Corporation and Dr. AK is a paid employee of Green Consultant Co. Ltd. They have no stocks or shares in both companies and they do not gain or lose financially through publication; consultation fees or other forms of remuneration from organizations that may gain or lose financially; patents or patent applications whose value may be affected by publication. There are no patents, products in development or marketing products to declare. This does not alter our adherence to PLOS ONE policies on sharing data and materials.

**Abbreviations:** TQOM, Trace quantities of organic matter; SPNES, Sodium polyoxyethylene nonyl phenyl ether sulfate; CT, Micro-focus CT scanner; TSM, Total suspended particles; DBP, Di-butyl phthalate; DEP, Diesel exhaust particulate; DEHP, Di-(2-ethylhexy) phthalate; TMPDIB, 2,2,4-trimethyl-1-1,3 pentanediol di-isobutylate; 2E1H, 2-ethyl-1-hexanol; DD value, Degree of deterioration in sample.

## 1. Introduction

The resolution of causes for damage in cement concrete structures is an important issue for the useful life of structures. To investigate the damage in concrete, a phenolphthalein solution is commonly applied to a cross section of the concrete and the extent of deterioration of the concrete is evaluated by the color corresponding to the reaction products, i.e. the value of pH (amount of hydroxide ions: $OH^-$). However, it seems insufficient to evaluate the damage to concrete by only the pH value observed on the concrete surface. The evaluation method for damage to modern concrete is based on the crack width (0.2 mm or more) on the concrete surface and the application of phenolphthalein. However, little research has been done on the degree of concrete damage and the relationship between cracks and deterioration. For this reason, it is not possible to evaluate the repair time and repair depth of concrete structures that had been thought to be damaged at present.

For this reason, there is a need to develop a new method for evaluating damaged samples that involve several chemical components in the concrete. Further, cement concrete structures also show unexplained damage that far exceed those caused by carbon dioxide in indoor tests.

This study suggests that modern evaluation methods to determine the damage to concrete are of limited effectiveness. Additionally, modern concrete structures have displayed a number of apparently incomprehensible and unexpected phenomena including delamination appearing near the surface, disaggregation (concrete changed to very fine particles), amorphization (crystal components in cement changed to amorphous substance), cracks, and pop-out of aggregate [1]. The disaggregation and peel-off of surface courses also occur on asphalt pavements in airport runways and highways [2], where the mechanisms have remained unexplained so far. Disaggregation in concrete bridges have also occurred at the interface between asphalt pavements and concrete slabs in concrete bridges. This damage leads to a shortening of the useful life of concrete structures and asphalt pavements, and also affects the safety of these structures. For this reason, this resolution of causes for damage and the development of new evaluation methods for the damage due to deterioration are urgent issues that are very important from the standpoint of structural safety.

In this study, we investigated a hypothesis that some of this unexplained damage in concrete could be caused by small amounts of unidentified organic matter. This hypothesis was based on the strange odor emitted by commercially available cement when it is mixed with water. To further elucidate this, this study presumes that TQOM not so far identified in fresh cement and present in the air has an effect on this strange damage to concrete structures and asphalt pavements.

To investigate this, many samples were taken from damaged concrete structures and damaged asphalt pavements where various unexplained phenomena had occurred. In order to investigate these causes, also unknown substances that have not yet been determined were sought from these, and the relationship between these causes and the strange phenomena was also investigated.

From the results obtained in the various experiments, it was concluded that all of the strange damage suffered by both kinds of structures were caused by TQOM and AE water reducing agent in air and/or cement, and surfactant in snow melting agents.

In this study, a new evaluation method was proposed for damage to concrete structures and asphalt pavements. Further, new evaluation methods for damage to concrete structures were also proposed using micro focus CT scanner (CT).

## 2. Experiments and test methods

It was considered that damages such as disaggregation, delamination, amorphization and cracks in concrete structures and asphalt pavements depend upon the materials (aggregate,

asphalt), type and content of TQOM in the air and/in the fresh cement, construction method, construction machinery employed, environmental conditions such as temperature and relative humidity, and surfactants in snow melting agents.

For the present study, we selected the following 11 types of themes to clarify the causes and mechanisms of damages (disaggregation, delamination, amorphization and cracks) in concrete structures and asphalt pavements, and many tests were performed. 1.Determination of details of the strange odor due to TQOM in fresh cement during mortar preparation (quality of cement), 2. Volume expansion of mortar samples due to multiple chemical components of TQOM in cement (TQOM, quality of cement, AE water reducing agent), 3. Delamination of mortar samples due to separation of TQOM (construction method and vibration processed), 4. Amorphization of mortar due to TQOM in fresh cement (TQOM, quality of cement, construction method) 5. Comparison of damage (crack widths and crack lengths) to modern mortar samples and a 120-year-old sample (quality of cement, TQOM), 6. Deterioration of mortar samples due to multiple chemical components of TQOM in fresh cement (quality of cement, construction method) 7. Pop-out of aggregate due to TQOM in cement and surfactants in snow melting agent (aggregate, concrete structure, surfactant of snow melting agent), 8. Organic matter in total suspended matter (TSM) (environmental condition) and emission sources (air pollution, environmental condition), 9. Content of water penetrated through water proofing sheets in cored samples from a new concrete bridge after transient moisture permeation tests (described in section 2.2.5) (structure, material, environmental condition), 10. Emission source and chemical components of the organic matter in damaged concrete structures and damaged asphalt pavements (material, TQOM in the air, multiple emission sources), 11. Cause of disaggregation in disaggregated concrete structures and disaggregated asphalt pavements (material, structure, construction method).

It was considered that it is necessary to understand these themes in disaggregation in concrete structures and asphalt pavements as will be described in section 3 (results and discussion).

## 2.1 Samples

**2.1.1 Asphalt, asphalt mixtures and asphalt pavements.** The asphalt sample was the penetration grade of asphalt (80/100) which is widely used for asphalt pavements in Hokkaido, Japan. It was used to investigate the emission source of organic matter in damaged concrete structures.

Asphalt mixtures on a newly constructed concrete bridge were taken from a two layered-pavement (surface course: dense graded mixture, 40 mm; base course mixture: porous mixture, 35 mm) with a water proofing sheet. These asphalt pavements were placed on a concrete bridge. In the asphalt runway of Nagoya airport, Japan, the asphalt pavement was placed as overlaid-pavements, two times (1960,1998) on an older concrete runway. The first overlaid-pavement was a coarse graded base course (17 cm), dense graded surface course (4 cm) in 1960, and the second overlaid-pavement had a coarse graded base course (17 cm), dense graded surface course (4 cm) added in 1998. Disaggregation had occurred in 2000 at the depth of 15 cm in the asphalt runway in Nagoya. The asphalt cored samples were collected from the disaggregated runway in Nagoya in 2000.

The asphalt pavement in the highway in service was formed by two layered-asphalt pavements (surface course: 5 cm; base course: 7 cm). The asphalt core sample was collected from the highway in service.

**2.1.2 Cement, mortar samples and concrete structures crushed in an earthquake.** Eleven types of cement from seven countries were collected to investigate the quality of the

cement to susceptibility to damage of mortar samples and used to investigate the degree of deterioration in new mortar samples. Nittetsu Portland cement (Nittetsu cement) in Japan was used as a standard modern cement, and the contents and chemical components of TQOM in fresh cement were investigated in this study.

The 120-year-old sample (120 year-sample in the following) is considered to have had a relatively long useful life as a mortar sample that was made 120 years ago in Otaru, Hokkaido, Japan and maintained indoors at room temperature for the 120 years. The composition and shape of the 120 year-sample are as follows. Asano Ordinary Portland Cement was used for the mortar sample, and the size of the mortar sample was width 4.5 cm, length 8 cm, thickness 2.2 cm, gourd type, and the composition of the sample was cement: volcanic ash: sand = 1: 0.5: 3, ratio of water/cement: 44%. The 120 year-sample was made from old cement (silica, clay, limestone) and AE water reducing agent was not used in this sample. This sample used natural sand and volcanic ash.

The composition and size of the new mortar samples are 2 x 2 x 2 cm, cement: sand = 1: 2, water cement ratio: 50%. All new mortar samples used Toyoura standard sand (high silica content, maximum particle size: 0.3 mm, coefficient of uniformity: 1.71) and AE water reducing agent (Pozzolith 70; 0.25% x cement).

Mortar samples were used to investigate the degree of deterioration due to TQOM and AE water reducing agent, amorphization, delamination, cracks, and volume expansion in the mortar samples. The TQOM in the crushed 120 year-sample was also investigated.

In this study, two types of structures: concrete bridge and runway pavements were used to evaluate the damage to concrete structures. The newly constructed concrete bridge was made with two layered-asphalt pavements with a water proofing layer and concrete slab (thickness: 16 cm; compressive strength: 24 MPa; the AE water reducing agent: 1% of cement; water/cement: 55%). The old concrete bridge (Tokachi bridge: slab: 30 cm, two layered-asphalt pavements: 7 cm) was made of old cement in 1934 and demolished in 1980. The cement was manufactured with no waste materials and no AE water reducing agent. The samples of the foundation (width: 40 cm, height: 45 cm) of the hand rail and bridge pier in the Tokachi bridge were selected from the surface (0-3cm) of core samples (diameter 10cm). The disaggregated concrete protective wall (wall height: 1 m, width 20 cm, wall bed: height 20 cm, width 50 cm), where wall and wall bed were held together by reinforced concrete, located at the median strip in a Kobe highway (elevated highway, disaggregated concrete from the Kobe highway in the following) and it was constructed thirty years ago. Samples of disaggregated concrete (disaggregated concrete wall) from the Kobe highway were collected for this study.

Runway concrete pavement (30 cm) from Chitose Airport, Japan was damaged by snow melting agent. In this runway, pop-out of aggregate and disaggregation of concrete were observed. Composition of the concrete in the concrete runway is as follows. Nittetsu cement, AE water reducing agent (Pozzolith 70; 0.25% of cement), W/C = 42%, Slump: 2.5 cm. Damaged concrete pavement (30 cm) of runway concrete divided into 9 layers, and volume expansion of mortar and the damage to the aggregate due to the surfactant of the snow melting agent were also investigated using these samples.

Many samples of damaged concrete were collected in Japan and Europe.

**2.1.3 TSM, tire debris, diesel exhaust particulate (DEP), and SPNES.** These materials were used to identify the emission sources in disaggregated concrete structures and disaggregated asphalt pavements.

As a tire, a new radial tire (Bridgestone: V-STEEL RIB294 245/70 R19.5, 1987) was used as a standard summer tire in Japan, and the surface of this tire was finely scraped and ground to obtain tire debris. The diesel exhaust particulate (DEP) was obtained with a dilution tunnel (Horiba, DLT-24150W) of the Japan Automobile Research Institute (JARI) in Tsukuba, Japan.

TSM was collected by a High-volume sampler (HC-1000N, Shibata Science Technology Ltd., Pallflex 2500 QAT-UP, quartz fiber filter, 23.5 × 17.5 cm, 1000 L/min) in Sapporo in summer (eight days in July 1998). The SPNES was used to examine the damage to concrete structures and asphalt pavements. It is used as a standard surfactant in windshield washer fluid throughout the world.

## 2.2 Analysis methods

In this study, the following tests were used to investigate the emission sources, water content, content and chemical components of the organic matter in damaged concrete structures and asphalt pavements, chemical components of the TQOM, and three-dimensional CT images were also used to investigate the crack distribution, amorphization, deterioration, delamination, amorphization, disaggregation in the damaged concrete and asphalt pavements.

**2.2.1 $^1$H NMR.** To measure the chemical peaks in spectra due to Nuclear Magnetic Resonance for organic matter, organic matter was in the Nuclear Magnetic Resonance device ($^1$H NMR: JOEL EX 400) and the device of $^1$H NMR will show peculiar peaks in the spectrum ($^1$H NMR spectrum) for the organic matter that is part of the chemical components of the organic matter sample, and peculiar peaks of the organic matter appears in the spectrum. The approximate estimation of organic substances can be made from these peak values.

Organic matter is extracted from the crushed material with Soxhlet and chloroform solution, and $^1$H NMR test is performed on the extracted organic matter. Subsequent $^1$H NMR Test was performed on this extracted material. (A. Moriyoshi, Journal of the Japan Petroleum Institute, 45(2) (2002) 84–88 [3]).

The $^1$H NMR test was performed to all samples including TSM, DEP, tire debris, SPNES, damaged deteriorated concrete, bitumen, and asphalt mixtures where blistering and disaggregation had occurred on the asphalt paved-runway in Nagoya airport. $^1$H NMR test was used to estimate the emission sources and the organic matter of each organic matter sample.

**2.2.2 GC-MS (JMS-AX-500).** Gas Chromatography-Mass Spectrometry (GC-MS) tests were used to identify organic matter and contents of chemical components mainly for organic matter extracted with the Soxhlet extractor and chloroform solution in the various samples such as TQOM in cement and the 120 year-sample, damaged deteriorated concrete, TSM, DEP, tire debris, bitumen. and others. (T. Tomoto, Constr. Build. Mater. 25(1) (2011) 267–281 [4]).

**2.2.3 HPLC.** High Performance Liquid Chromatography (HPLC) Tests were used to identify organic substances in heavy oil such as bitumen, disaggregated concrete from the Kobe highway, tire debris, DEP, and TSM. (T. Tomoto, Building and Environment 44 (2009) 2000–2005 [5]). The HPLC datum used the results reported in published papers [5].

**2.2.4 High volume sampler.** The high-volume sampler test was used to collect the TSM in the air as follows. (T. Tomoto, Building and Environment 44 (2009) 2000–2005 [5]).

The organic matter identified by high volume sampler was used for identification of emission sources and chemical components of organic matters for the disaggregated concrete in the Kobe highway obtained from Kobe city. This result was evaluated using data in published papers using the same GC-MS and HPLC apparatus in section 2.2.3. (T. Tomoto, 44 (2009) 2000–2005 3]).

**2.2.5 One-dimensional transient moisture permeation test.** The one-dimensional transient moisture permeation (transient moisture permeation) apparatus is a device newly developed in our laboratory in Hokkaido University, Sapporo, Japan and manufactured for this study to measure the moisture by the weight changes (±0.1g) of samples under respiratory action of concrete structures and asphalt pavements. In this test, a KCL-1000 apparatus

(Tokyo Rika Co. Ltd.,) was used [6]. (I. Sasaki, Journal of the Japan Petroleum Institute 49 (1) (2006) 33–37 [6]).

As a result, this equipment can reproduce the summer environmental conditions (temperature and relative humidity of summer condition: 24 hours: one day) in the laboratory [6] like that giving rise to the peel-off phenomena (length: 8 m, width: 4 m, thickness: 5 cm) of the surface course that occurred on the runway (2740 x 45 m) of the asphalt pavement at Nagoya Airport investigated here. The peel-off phenomena occurred at 15:20, 2nd July, 2000, and the environmental conditions (humidity, temperature) were reproduced in the laboratory (summer condition) for the 24 hours of the 2nd July of Nagoya runway.

**2.2.6 Micro focus CT scanner (CT).** This test was used to identify cracks (crack widths and crack lengths) and amorphous substances in aggregate and concrete samples (T. Tomoto, Constr. and Build. Mater. 25 (1) 267–281 [4]).

The larger the number of pixels in the CT apparatus (Shimadzu Corporation, inspeXio, SMX-22CT), the more accurate the shape of the sample can be reconstructed [7]. Here, a 512-pixel device was used for ordinary CT imaging, while a 1024-pixel device was used to observe the white fine particles (disaggregation, diameter: 0.01–0.3 mm) in the sample.

**2.2.7 Software for three-dimensional (3D) crack analysis and DD values.** his software was used to identify cracks (crack widths and crack lengths), amorphous substances in aggregate and concrete samples. Software for three-dimensional crack analysis (VG StudioMAX 2.0 and Exfact Analysis for Porous/Particles 2.0, NVS Co. Ltd., Tokyo) to examine the crack properties in concrete and aggregate from CT images. The degree of deterioration (DD value: it was described later) of samples was analyzed by two-dimensional CT images and Image-Pro Analyzer 7.1 (Nippon Rover Co. Ltd., Tokyo, Japan).

## 3. Results and discussion

The Table of 'Results and Discussion' is as follows.

3.1 Organic matter in cement 3.2 Expansion of new mortar samples and pop-out of aggregate 3.3 Delamination 3.4 Amorphous substance in mortar samples 3.5 Damage to new mortar samples and 120 year-sample 3.6 Deterioration in mortar samples (DD value) 3.7 Pop-out of aggregate in concrete pavements 3.8 Trace quantities of organic matter (TQOM) 3.9 Respiratory action of concrete structures and asphalt pavements 3.10 Organic matter in damaged concrete structures and damaged asphalt pavements 3.11 Disaggregation in asphalt pavements and concrete structures.

## 3.1 Organic matter in cement

The organic matters in Nittetsu cement and crushed 120 year-sample were extracted 4 times (24 hours) using Soxhlet extractor and chloroform solution. We call it 'substance extracted with Soxhlet'.

The following substances were identified as TQOM in the cement using GC-MS. For the Nittetsu cement in 10 g of cement: 119 μg of Di-butyl phthalate (DBP) and 2.1 μg of Di-(2-ethyhexyl) phthalate (DEHP) were detected. In the 120 year-sample in 10 g of Asano Portland cement: 1 μg of DBP, 0.1 μg of DEHP, and trace amounts of 2, 2, 4-trimethyl-1, 3-pentanediol di-isobutylate (TMPDIB: Texanol) were detected. These substances in 120 year-sample are TQOM in air. It was considered that the organic matter in the 120 year-sample was absorbed from the air. The above finding suggests that the strange odors emitted from cement involved TQOM such as phthalates (DBP, DEHP) when brought in contact with water are possibly due to chemical hydrolysis of phthalates in the cement and that the organic substances contained in cement were hydrolyzed instantaneously and generated the odor of 2-ethyl-

                                                    

1-hexanol (2E1H) gas, and/or butanol. The odor was noted to occur instantaneously during the preparation of the mortar samples using the 9 types of cement but not with the extracted cement (cement extracted organic matter from Nittetsu cement with Soxhlet and chloroform solution), indicating that trace amounts of phthalates were contained in these cements. It has been believed that the TQOM in cement has no effect on the damage to concrete structures and asphalt pavements. A small amount of phosphate compounds (Nittetsu cement), which is not dissolved in chloroform solution and could not be extracted by this method using Soxhlet extractor, was detected at a concentration of 0.12% by ICP (emission spectral analysis: ALS Chemex Ltd., Canada). These series of experiments suggest that commercially available Nittetsu cement contains phthalates and phosphate compounds, and that modern concrete structures are also exposed to trace amounts of surfactants added to snow melting agents. The effect of surfactants on the damage induced deterioration of concrete had been also investigated using cement paste samples (mixtures of cement and water)(T. Tomoto, Constr. Build. Mater. 25 (1) (2011) 267–281 [4]). In this test, cement paste instantaneously reacts with the chemical components in cement in aqueous solutions of trace amounts of surfactants (anionic and non-ionic) to produce water-soluble calcium and refractory calcium. In addition, it is known that very small amounts of phosphate compounds contained in meat-and-bone meal in cement reacts with calcium components in aqueous solutions [8,9]. Based on the above, it appears possible that these organic components (TQOM) involved with the phosphate compounds in cements affect the unexplained damage and deterioration of concrete.

### 3.2 Expansion of new mortar samples and pop-out of aggregate

It was considered that the cause of pop-out of aggregate was caused by volume expansion of aggregate and mortar in concrete and the causes of volume expansion were TQOM in cement and surfactant in snow melting agents. To substantiate this, volume expansion and delamination of new mortar samples (at 30 days after placement) were measured mainly to investigate the volume expansion at the side views of center sections of mortar samples (2 x 2 x 2 cm) in two-dimensional CT images (512 pixel).

To investigate the effect of the type of cement and TQOM in cements on the volume expansion of the mortar samples, mortar samples made of 11 types of cements were prepared in this study.

Fig 1 shows volume expansion of four samples (samples A, B, C, and D) at 30 days after placement with the red lines on the samples showing the original height of the sample. White

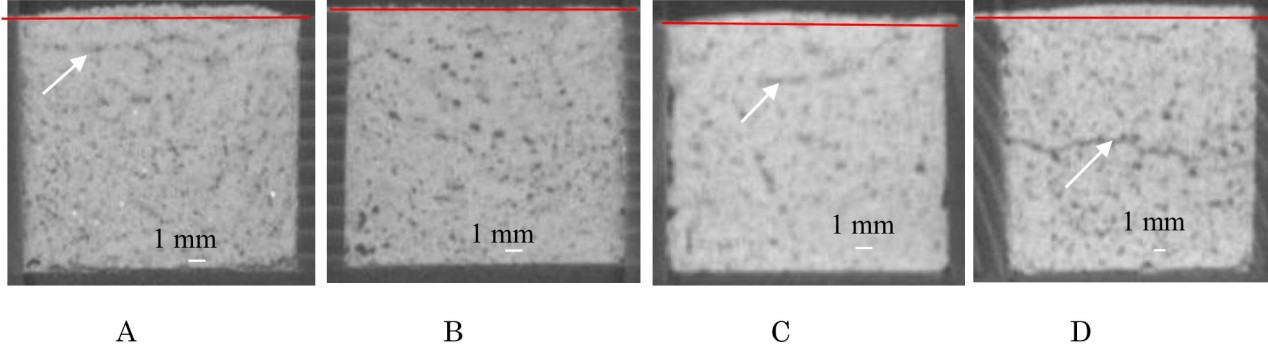

| A | B | C | D |

**Fig 1. Volume expansion and delamination (cracks: White arrow) of new mortar samples.** A Volume expansion of sample A (Nittetsu cement; 6%). B Volume expansion of sample B (extracted Nittetsu cement; 0%). C Volume expansion of sample C (extracted cement including AE water-reducing agent; 9.4%). D. Volume expansion of sample E (Kazakhstan cement including AE water reducing agent; 6.1%).

arrows in Fig 1 show the delamination (cracks) in the samples. The properties of these sample are described next.

The chemical components and the contents of organic matter in these samples are described in section 3.6.

Sample A contained phthalates and phosphate compounds, sample B contained phosphate compounds, and sample C contained phosphate compounds and AE water reducing agent.

The organic matter in sample D (Kazakhstan cement) is not known.

Delamination (white arrow) in sample B (extracted Nittetsu cement) was not observed.

In order to investigate when volume expansion in mortar samples will occur, the data of two-dimensional CT images at 1, 3, 8, 14, 21, and 30 days after placement were also examined using these samples. The volume expansion at 1 day in four samples (sample A, B, C, and D) was very similar to the volume expansion at 30 days.

Five kinds of cement (normal, early strength, blast furnace) in Japan were collected for cement paste, and these samples were tested by the Double Cylinder Chamber method (30˚C, relative humidity 50%, at 450 ml/min) using powdered cement paste sample of 10 g and DEHP (0.4 g) to measure the emission rate of 2E1H gas (T. Tomoto, Building and Environment 44 (2009) 2000–2005 [5]). With this device, the emission of 2E1H gas was observed in the sample (Nittetsu cement +AE water reducing agent; 0.25% x cement; 29.9 mg/h/m$^2$) containing the AE water reducing agent at one day, was five times of the emission level of the cement (Nittetsu cement, 5.7 mg/h/m$^2$) without AE water reducing agent. This gas of 2E1H has been reported to release a large amount (20–40 mg/h/m$^2$) over more than 3 months in this test in all cement paste samples (T. Tomoto, Building and Environment 44 (2009), 2000–2005 [5]).

This result shows that the AE water reducing agent related both to the damage in concrete as well as to the generation of 2E1H gas (volume expansion during compaction of sample).

This generation of 2E1H gas is considered to be related to the volume expansion of concrete when it is placed.

It strongly suggests that volume expansion in mortar samples was mainly caused by phthalates and AE water reducing agent and that cracks occurred in the mortar samples due to volume expansion. It seems that these chemical reactions (volume expansion) due to TQOM in the mortar samples occurred quickly by the gas, which generated by hydrolysis of chemical components in cement due to phthalates and AE water reducing agent in the mortar samples, leading to the volume expansion in mortar samples.

The effects of AE water reducing agent and phthalates in the cement (Nittetsu cement) on volume expansion in mortar samples were investigated using the volume expansion of mortar in Table 1. Sample B (phosphate compounds) is the sample where phthalates in cement were extracted using chloroform and Soxhlet extractor and there was no volume expansion in sample B. Expansion of mortar sample involved phthalates and phosphate compounds in cement was observed in sample A. The volume expansions were 6% in sample A, 0% in sample B, respectively.

Table 1 shows the degree of volume expansion and DD values (it is further described later) for mortar samples (2 x 2 x 2 cm) made of the 11 types of cement. Here, the extracted Nittetsu cement shows cement with TQOM extracted from Nittetsu cement using chloroform and Soxhlet extractor.

The chemical contents in the Kazakhstan cement are unknown. Emission rate of 2E1H (volume expansion) depends upon the type (Normal, High-early strength, Blast furnace) of cement (T. Tomoto, Building and Environment, 44 (2009) 2000–2005 [5]). Sample E (Fig 1D) including AE water reducing agent was excluded in this study. However, the side surfaces of both samples (samples C and E) with AE water reducing agent were much smoother than

 

**Table 1. The degree of deterioration (DD value: %) in mortar samples, content of DBP in cement and /or mortar, and maximum expansion of samples (%).**

| | Sample No. | A | B | C | D | E | F |
|---|---|---|---|---|---|---|---|
| | Company | Nittetsu | Extract (N) | Extract(N)+AE* | Kazakhstan(K) | (K)+AE* | Holcim |
| | Country | Japan | Japan | Japan | Kazakhstan | Kazakhstan | Germany |
| | | JIS R 5210 | JIS R 5210 | JIS R 5210 | PC400- D20 | PC400-D20 | CEM II/B-T |
| DD value: Normal Portland Cement (%) | | **8** | **4.9** | **9.4** | **20.4** | **16.5** | **8.1** |
| DD value Portland-Composite Cement (%) | | | | | | | |
| Content of phthalate compounds | | | | | | | |
| DBP: µg/10 g of cement | | 119 | 3 | 3 | N/A | N/A | N/A |
| Ratio of cement (%) | | 0.0012 | 0.0003 | 0.0003 | N/A | N/A | N/A |
| Ratio of mortar (%) | | 0.00033 | 0.0001 | 0.0001 | N/A | N/A | N/A |
| Max. expansion of Sample (%)** | | 6.8 | 0 | 4.2 | 3.3 | 6.1 | 3.3 |
| | Sample No. | G | H | I | J | K | 120year |
| | Company | Italcementi | UNACEM | Cementi | DalianOnoda | CalPortland | Asano |
| | Country | Italy | Peru (PM) | Italy | China | USA MOJAVE | Japan |
| | | with CaCO$_3$ | ASTM: C-150 | CEMII/B-M | ASTM Type I | Type II/V | 120year |
| DD value: Normal Portland Cement (%) | | **13.4** | **10.6** | **10.2** | **10** | **4.9** | 1.8 |
| DD value Portland-Composite Cement (%) | | | | | | | |
| Content of phthalate compounds | | | | | | | |
| DBP: µg/10 g of cement | | N/A | N/A | N/A | N/A | N/A | 1 |
| Ratio of cement (%) | | N/A | N/A | N/A | N/A | N/A | N/A |
| Ratio of mortar (%) | | N/A | N/A | N/A | N/A | N/A | N/A |
| Max. expansion of sample (%)** | | 1.1 | 1 | 2 | 1 | 1.8 | N/A |

*: AE water-reducing agent: Pozzolith No.70 (lignin sulfate and polyol compounds type, content: 0.25% x cement).

**: Maximum volume expansion of Sample (2 x 2 x 2 cm) at 30 days after placement.

**Bold**: Larger than the degree of decalcification in 120 year-sample.

sample A without AE water reducing agent. They are shown in Fig 1C and 1D. It is considered that the uneven (fine irregularities) surfaces in samples A and B were caused by low viscous substances in the samples, but the smooth surfaces on samples C and E were caused by high viscous material in the samples.

The side surface of samples after volume expansion of sample A (Nittetsu cement, Fig 1A), sample C (extracted cement + AE water reducing agent, Fig 1C) and sample E (Kazakhstan + A E water reducing agent, Fig 1D) showed a slightly convex shape.

Although the number of samples is small, the results suggest that when a sample contains phthalates, and/or when AE water reducing agent is added, the mortar sample expands quickly during 1 day of curing. It may be that mortar samples are prone to swelling (volume expansion) during hardening when phthalates in fresh cement and AE water reducing agent is present in the mortar samples.

## 3.3 Delamination

Delamination (cracks) occurred near the surface of concrete structures, and it has been an unexplained phenomenon in concrete structures. It was postulated that the cause of the delamination was separation due to TQOM such as phthalates and/or AE water reducing agent in cement from the vibration during compaction and the construction methods during the preparation of mortar samples. In this study, first, effects of TQOM in cement and AE water reducing agent on the delamination in the four mortar samples (samples A, B, C, and E) was

examined using two-dimensional CT images. These samples are shown in section 3.2 with the images of two-dimensional CT (512 pixel) for the samples at 30 days after placement. In Fig 1, two types of damage (dark spot and dark band) are shown. These substances were amorphous materials, and the damage occurred by different sources of damage due to TQOM in samples. Dark band (thickness: 0.4 mm or narrower) distributed in a plane in the samples, the dark spots (diameter: 0.4 mm) were independent damage in samples. We found the dark band (plane) taken with the 512-pixel CT represents a crack [10]. The dark band in sample C in Fig 1C (extracted cement + AE water reducing agent) shows it as relatively thick. The relation between dark spots and dark band (cracks) will be described in section 3.5. Fig 1 shows that the black dots were extremely few up to about 2 mm from the side surface. It suggests that separation of TQOM and AE water reducing agent in samples occurred in these areas of the mortar samples. The dark band changed to thicker, widening with longer curing of the mortar sample (from 1 day to 30 days) and connected with the dark dots in the delamination shown by the white arrows in Fig 1A, 1C and 1D. It shows that the crack width changed with time elapsed, because dark dots are the deterioration in samples and the dark band shows the cracks in samples (A. Moriyoshi, Hokudai HUSCAP LETTER, (2014) 26 [10]). Dark bands of the image in the 512-pixel CT shows the crack in the image in the 1024-pixel CT in the same sample. It shows that wider band has cracks in images in the 512-pixel CT, and the darker the color of the band, the wider the crack.

Further, crack width of the delamination is not constant, the crack width in delamination change in crack (it will be described in section 3.7(pop-out)).

In section 3.7, the dark band in Fig 1B changed due to addition of AE water reducing agent to the thicker bands such as in Fig 1C and 1D. Delamination occurred in the horizontal direction at the side view in Fig 1A, 1C and 1D. It shows as white arrows that the delamination in samples was caused by the construction method and vibration during compaction.

The amount of DBP and/or DEHP was not measured in all samples, but small amounts of phthalates and AE water reducing agent were thought to be the cause of the delamination cracks.

Further, the delamination (cracking) may be related to DD values (Table 1) which are expressed as the degree of deterioration in section 3.6 (DD value). The delamination due to TQOM in mortar samples is described in the following section 3.4 (amorphization).

## 3.4 Amorphous substance in mortar samples

It was considered that amorphization in cement concrete occurred by chemical changes in the main components (CaO, $SiO_2$) in the cement and due to the TQOM in the cement. It was considered that amorphous substances were caused by TQOM in the cement and AE water reducing agent which may be unevenly distributed in the mortar samples during compaction of mortar samples, referring to the crack pattern of the samples near the surface in the damaged concrete runway in section 3.7 (Pop-out). The materials of new mortar samples used in this study were made of the same materials (Nittetsu cement, AE water reducing agent; Pozzolith 70, 0.25% x cement) as those in the damaged concrete runway in Chitose, Japan (it will be described in section 3.7 (pop-out)). New mortar samples were compacted by steel bars and vibration. Further, phthalates in Nittetsu cement are little soluble in water, and DBP, DEHP and AE water reducing agent (TQOM) in new mortar samples have different specific gravity and water solubility of 1.043 and 10 mg/L for DBP, 0.9801 and 0.25 mg/L for DEHP, and 1.04–1.06 and they are water soluble with AE water reducing agent. Making it likely that some components of these chemical substances (TQOM) separated in water by the vibration and construction method (two layered-construction of mortar sample) during placement of the

   

mortar sample and/or by the differences in specific gravity and water solubility. The properties (DD value and volume expansion) of new mortar samples of the 11 types (2 x 2 x 2 cm) are shown in Table 1 in Section 3.2, for the mortar samples of two-layered mortar (1 + 1 cm: construction method). The DD value shows the degree of deterioration in mortar samples which will be described in section 3.6. Fig 1 shows that the AE water reducing agent, (specific gravity: 1.04–1.08) would concentrate mainly at the bottom (construction joint) of the second mortar layer (depth from surface: 1 cm) of sample E in Fig 1D (included AE water reducing agent; specific gravity; 1.04–1.08), while the DEHP (specific gravity; 0.98) would concentrate at the near surface of sample A in Fig 1A (it included DEHP, DBP, specific gravity of DEHP; 0.98, and specific gravity of DBP; 1.05. This deterioration due to these phthalates and AE water reducing agent are shown as black dots in Fig 1.

The side view of the delamination (crack: white arrows) in sample C in Fig 1C (extracted cement + contained AE water reducing agent with specific gravity 1.04–1.08) located at a relatively lower part than that of sample A in Fig 1A (Nittetsu cement included phthalates such as DBP and DEHP). In this location it is suggested that a part of these phthalates (DBP, DEHP) and AE water reducing agent in new mortar samples separated in water by vibration during compaction of the samples concentrated at the construction joints (depth from surface: 1 cm) and near the surface of new mortar samples (2 x 2 x 2 cm). Since the phthalates in cement is poorly soluble in water, it may be assumed that they react gradually chemically with the calcium component in the cement to generate amorphous substances, and suggest that this leads to the delamination and disaggregation through the amorphization of mortar samples.

Further, to investigate the damage such as disaggregation, crack and amorphization in the new mortar samples, amorphous substances in the mortar samples were examined using two-dimensional CT images. Fig 2 shows two-dimensional CT images (1024 pixel) 30 days after the sample creation at the surface, at a depth of 10 mm into the mortar samples.

Fig 2 shows that the disaggregation appears as shown out-side the fine particles (tips of white arrows) in the amorphous substances (black).

Fig 2A–2D. show the two-dimensional CT images on the surface of samples at the center of samples 30 days after its placement. The white fine particles are independent and the outside of the white fine particles show as black. It shows that disaggregation occurred at the outside of the white fine particles in the samples.

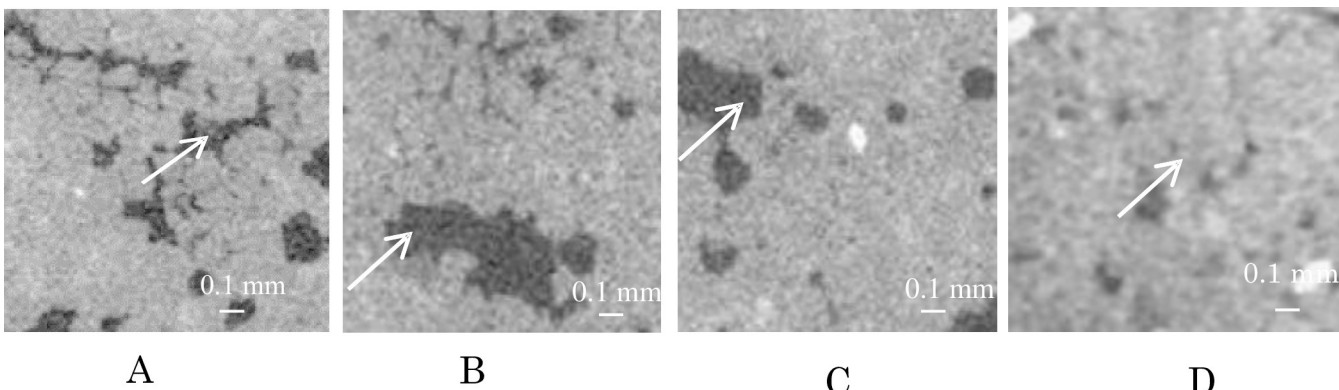

**Fig 2. Amorphous substances (black) and disaggregation in amorphous substances in new mortar samples.** A. Amorphous substance of sample A (Nittetsu cement). B. Amorphous substance of sample B (Extracted cement). C. Amorphous substance of sample C (Extracted cement + AE water reducing agent). D. Amorphous substance of 120 year-sample.

Disaggregation showed black in the CT images, but it does not show real black like in voids. It is thought that the materials on the outside of the fine particles changed to substances with a low specific gravity, and the substances outside the fine particles in the amorphous material were very finely pulverized, causing the disaggregation.

These results suggest that TQOM in the mortar samples affected the deteriorations of the samples and that TQOM formed amorphous substances such as the black dots and black bands (cracks) in these samples.

Further, disaggregation will be described in detail in the following section.

Fig 2 shows two-dimensional CT images (1024 pixel) taken 30 days after sample preparation at the surface, at a depth of 10 mm in the samples. All samples show amorphous substances in these CT images. The dark amorphous substances in the CT images are not fully black like voids (air), and do not represent a cavity in the sample, but rather appears to show some substance. It is known that when cement concrete deteriorates, multiple cement components, like C-S-H (CaO·ySiO2·zH2O, x/y = 1.2–2.0, z/y≒1.59, CaO, and SiO2 are crystalline) change to amorphous substances [11]. Tomoto reported that cement paste samples (10 g) emitted 2E1H gas (20–40 mg/h/m$^2$) which was hydrolyzed by DEHP (0.4 g) in a double-cylinder-chamber method (30˚C, relative humidity 50%, clean humid air 450 ml/min.), and DEHP (0.4 g) is a poorly soluble substance (0.25 mg/L) in water, and the emission of 2E1H (20–40 mg/h/m$^2$) continued for three months or longer. This result shows trace quantities of organic matter such as TQOM of DEHP (phthalates) affected the damage (hydrolysis) of cement paste samples during long periods (T. Tomoto, building and environment, 44 (2009), 2000–2005 [5]). We found that the amorphous substance in the CT images has a somewhat smaller specific gravity and higher water content than that of normal mortar. We also found that the disaggregation of mortar had occurred due to amorphization of mortar, because the solid substances such as C-S-H (main chemical components in cement: CaO, and SiO$_2$) had been shown as black bands (samples A, B, C, and D in Fig 1) with connected black dots in the CT of 512-pixel CT images in Fig 1, but cracks appear in samples A and B (Fig 2A and 2B) in the 1024-pixel CT in Fig 2. Here, it shows that they changed to amorphous substances. However, it is difficult to determine the position of this substance (amorphous substance) in the sample using conventional equipment such as electron or light microscope. The CT device emits X-rays to samples that pass through the sample, and the concentration of these X-rays in the CT images is measured finely in three dimensions for each voxel (small volume) at the light receiving device and the images in the CT device expresses the concentration of transmitted X-rays at which X-rays pass through the sample appearing as a black image, the concentration of transmitted X-rays of substances containing water that exhibit smaller specific gravity and they show darker image than the concentration of transmitted X-rays of the part of regular mortar through the X-rays in the CT image and the voxel in the CT images is displayed as black. Disaggregation means that the main chemical components in cement mortar (CaO and SiO$_2$) are amorphized by TQOM. From this result, we found that the black substance in samples that forms a low-density area in the CT image and they are considered to be an amorphous substance containing multiple chemical components of cement mortar involved with water of lighter specific gravity. This evaluation method can be applied to evaluate the deterioration in the depth direction in mortar samples.

This study considers the amorphous substance to be an indicator of the deterioration of the mortar sample. It was considered that the larger the area of amorphous substance, the more the mortar sample had deteriorated.

The form of this amorphous substance varies from sample to sample, and the form of amorphous substance changes slightly depending on the type and amount of TQOM and AE water reducing agent.

### 3.5 Damage to new mortar samples and 120 year-sample

There is a controversy about concrete damage, disagreements about which is the main damage, cracks or deterioration in concrete structures.

To resolve the problem it has been desirable to develop a new method for evaluation of concrete structures. To assist in this evaluation, this study was attempted to determine which is the main damage (crack or deterioration) for new mortar samples made from recently manufactured cement and a 120 year-sample which has had a long life. The method was examined using the wider crack widths and lengths of the wider crack widths in both samples using three-dimensional crack analysis and CT scanner. When the crack width on the surface of concrete structures become 0.2 mm, it is necessary to repair the structures in Japan. But only crack width on surface of concrete is not sufficient to evaluate the damage to concrete structures, because crack width in delamination changes in cracks (it was described in 3.3).

We examined which of the seven factors is the most likely to inform us for an evaluation of crack damage of concrete structures.

The crack analyses of damage in mortar samples were performed using the software of three-dimensional crack analysis for CT images (NVS Co. Ltd.,: VG StudioMAX 2.0 and Exfact Analysis for Porous/Particles 2.0). The CT images for the 120 year-sample were taken by the 1024-pixel apparatus (voxel size: 0.138 x 0.138 x 0.138 mm) and new mortar sample A (Nittetsu cement) were taken by the 512-pixel apparatus (voxel size: 0.05 x 0.05 x 0.05 mm.

First, the three-dimensional crack distribution in both samples were examined.

Fig 3A and 3B show the crack width and crack length of the sample of 120 year-sample and mortar sample A. Fig 3 shows that the length of cracks (1 mm and/or longer) in sample A was longer than in the 120 year-sample.

Many narrower crack-widths of 0.1mm cracks (yellow) were observed in Fig 3A with many wider crack-widths of 0.138mm (red) was observed in Fig 3B.

The Fig 3B image shows the distribution of three dimensional-cracks in mortar sample A at 30 days after its formation and the crack width due to delamination (white arrows). The crack widths in the delamination in Fig 3B changed from 0.138 mm (red) to 0.414 mm (olive).

Further, the crack widths and crack lengths in both samples were examined to compare the crack damage (crack or deterioration) of both samples using following seven factors for crack damage. These results are listed in Table 2.

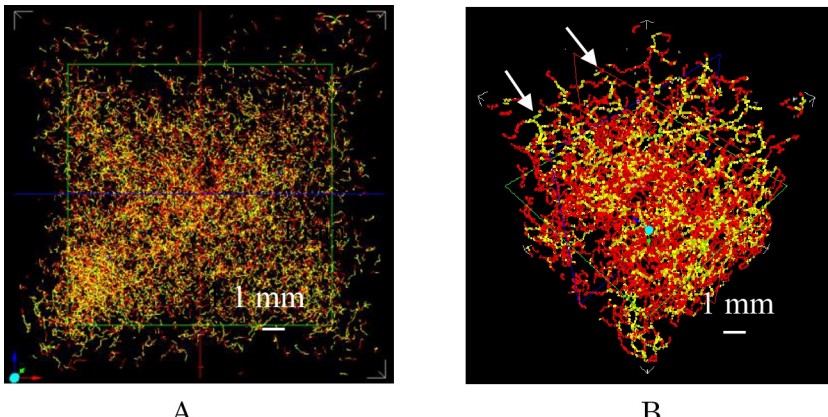

A                                                    B

**Fig 3. Distribution of three dimensional-cracks in new mortar sample A and the 120 year-sample.** A. Crack widths from 0.05 mm (red) to 0.25 mm (purple) in the 120 year-sample. B. Crack widths from 0.138 (red) to 0.552 mm (purple) in mortar sample A.

**Table 2. Properties of 120 year-sample and new mortar sample A (30 days after formation).**

| | 120 year-sample (1024 Pixel) | sample A (512 Pixel) |
|---|---|---|
| Porosity (%) | 3.25 | 14.2 |
| DD value (%) | 1.8 | 8 |
| (1) Ratio of DD value (RDD) | 1 | 4.4 |
| Distribution of void (mm³) | 0.0001~1.0 | 0.02~10 |
| Average of void (mm³) | 0.02 | 0.2 |
| (2) Crack width (CW) (mm) Color: | 0.05 (red: 41.83) | 0.138 (red: 67.6) |
| (3) Ratio to total crack | 0.1 (yellow: 48.06) | **0.276 (yellow: 30.8)** |
| Length (RTCL: %) | 0.15 (green: 9.44) | **0.414 (olive: 1.6)** |
| | 0.2 (blue: 0.66) | **0.552 (blue: 0.01)** |
| | **0.25 (purple: 0.0081)** | |
| (4) Sum of wide crack length over 0.276 mm (SWCL: %) | 0.0081 | 30.8+1.6 +0.01 = 32.41 |
| (5) Ratio of wide crack length (RWCL)* | 1 | 32.41/0.0081 = 4000 |
| Measured volume of sample (mm³)(V) | 15625 | 4574.3 |
| (6) Wide crack length over 0.276 mm (mm) (WCL) | 12.3 | 5463.5 |
| (7) Ratio of wide crack length over 0.276 mm measured volume (RWCL/V: %) | 0.078 | 119.4 |

**Bold**: crack over 0.276 mm.

Color: Three dimendional crack analysis (VG StudioMAX 2.0 and Exfact Analysis for Porous /Particles 2.0).

*: RWCL shows the ratio of wide crack length over 0.276 mm (WCL) in two samples.

Table 2 shows the crack properties (crack width), void properties and ratios of crack lengths of specific crack widths to the total crack length in the 120 year-sample and sample A. Table 2 shows the index of the main damage for mortar sample is the wider crack width and crack length of the wider crack width rather than deterioration (DD value) for damage evaluation of mortar samples.

(1) DD value and the ratio of the DD value of both samples (RDD), (2) Crack width (CW), (3) Ratio of crack length for a crack width (CW) to the total crack length (TCL) in the measured volume of the CT in the sample (RTCL), (4) Sum of wider crack lengths over 0.276 mm to the total crack length in samples (SWCL), (5) Ratio of wider crack lengths (RWCL), (6) Ratio of wider crack lengths (WCL) with a crack width of 0.276 mm or more to total crack length of the measured whole sample (RWCL), and (7) Ratio of wider crack lengths (WCL) with crack widths of 0.276 mm or more to the measured volume (V) by the CT (RWCL/V) (Table 2).

This study was considered that relatively large value and relatively large ratio for both samples in the seven factors are useful and necessary to evaluate the damage (crack and deterioration) of samples. The two factors for cracks were selected as follows. The factors of (4) the sum of the wide crack length over 0.276 mm to the total crack length in samples (SWCL), and (7) the ratio of the wide crack length over 0.276 mm to the total crack length in samples (RWCL/V) agree with this requirement, because (4) and (5) are the same value. The sum of the wide crack length over 0.276 mm (SWCL) for the 120 year-sample and sample A is 0.0081 and 32.41, respectively, and ratio of wide crack lengths over 0.276 mm/measured volume (RWCL) for the 120 year-sample and sample A is 0.078 and 119.4, respectively. Therefore, this may be assumed to show that two factors ((4) and (7)) to evaluate the damage of samples are useful with wider crack widths (0.276 mm over) and those crack length/volume in samples.

Chemical deterioration of concrete structures can be classified into two categories, cracks and deterioration (amorphization). In this study, deterioration in mortar samples was examined using the DD value in mortar samples. It will be described in section 3.6 (DD value).

The DD value of sample A was 8% and the DD value of the 120 year-sample was 1.8% in Table 1. Both these values were almost of the same order. It shows that the DD value does not show the main factor for crack damage in these samples and shows only deterioration. It would seem that the DD value is useful to express the deterioration in samples.

From the above results, comparing the crack damage of both samples (sample A and 120 year-sample), sample A has many cracks with a wider crack width (0.2 mm over) that needs repair, the cracks are long, and the DD value is also relatively large. Then, it seems that the evaluation method for crack damage in concrete structures is a more appropriate factor than the deterioration.

From this, it seems that mortar samples using modern cement A is liable to deteriorate for short times and it is more fragile and weaker, than the 120 year-sample.

## 3.6 Deterioration in mortar samples (DD value)

Amorphous substances show damage such as cracks and deterioration of concrete structures, but the degree and causes of the damage to in concrete structures have not be shown until this present study.

A new evaluation method and causes of deterioration (amorphous substances) in concrete structures were examined in this study using CT images. First, we calculated the ratio of the area of amorphous substances to the area of normal mortar in two-dimensional (2D) CT images as a DD value indicating the degree of deterioration of a new mortar sample (using 11 types of mortar) and a 120 year-sample. Table 1 shows the DD value for these mortar samples (11 types of mortar). In this analysis, all amorphous substances in bands and dots were included. But the amorphous substances (damage) in the band type of substances (crack) were few, because the crack thickness (amorphous substances, cracks) is very narrow. These narrow images of amorphous substances in CT images (1024 pixel) are shown in Fig 2.

The DD value varies from cement to cement. When the 11 types of cement here were individually mixed with water, all cements except that with the extracted cement generated a strange odor when the mortar was made. This strongly suggests that the strange odor generated at the time of mortar preparation likely originated from an unknown substance and it was thought to be a gas evolved by the hydrolyzing of phthalates. Further, to investigate the cause of origin of the amorphous substances determined in mortar samples, the following samples (sample A, B and C) with different DD values in Table 1 were selected. These DD values for the selected samples are as follows. Sample A (8%) is Nittetsu cement, sample B (4.9%) is extracted cement (cement extracted organic matter from Nittetsu cement), sample C (9.4%) is extracted cement involved with AE water reducing agent.

These DD values were somewhat larger than that of the 120 year-sample (1.8%). It suggests that the DD values depend upon the content and type of TQOM and AE water reducing agent in the cement.

Images (512 pixel of CT) of sample B made of cement extracted phthalates from the cement (Nittetsu cement) of sample A (included phthalates) with a Soxhlet extractor and chloroform showed almost no dark band (crack), compared to sample A without extraction. This indicates that sample A (phthalates) has more cracks than sample B (no phthalates).

Table 1 suggests that all new mortar samples deteriorated by TQOM and AE water reducing agent, as indicated by the DD value.

Since the DD value of the mortar samples appears to be related to the types of TQOM in the cement, the effects of TQOM (DBP) on the deterioration in the three mortar samples

(samples A, B, and C) was examined further. The content of DBP in sample A (Nittetsu cement) was larger than the content of DEHP.

The ratio of the content of TQOM such as phthalates, phosphate compounds and AE water reducing agent to mortar in Nittetsu cement were as follows. Phthalates: 0.0003% (cement x 0.0012%), Phosphate compounds: 0.034% (cement x 0.12%), AE water reducing agent: 0.07% (cement x 0.25%). Phosphate compounds, 0.12%, was contained in Nittetsu cement as detailed in section 3.1 (organic matter).

These data are described in section 3.1 (organic matter). This organic matter was contained in Nittetsu cement as TQOM and this small amount of phosphate compounds were also thought to be deteriorated to the mortar samples [12].

The effects of single organic substances in this organic matter (TQOM) in Nittetsu cement on the damage of new mortar samples were examined and it was calculated using the DD values of 3 types of mortar samples (sample A, B, and C). The DD value at 30 days after formation of the mortar were used in this calculation.

The contribution ratio for the damage to samples in the 3 types of mortar samples (samples A, B, and C) made of Nittetsu cement containing multiple organic matter components (TQOM) versus the deterioration (DD value) was determined using the DD value at 30 days after formation of the mortar. All three types of new mortar samples (sample A, B, and C) contained trace amounts of phosphate compounds (0.12%). This small phosphate compound content was also thought to be deteriorated to mortar samples.

It was assumed that the DD value indicating the deterioration of the new mortar samples (sample A, B, and C) represents the degree of deterioration caused by the trace amounts of three kinds of organic substances (TQOM: phthalates, phosphate compounds, and AE water reducing agent), because phosphate compounds could not be extracted by Soxhlet extractor and chloroform solution. The three types of mortar samples are as follows. Sample A was made of Nittetsu cement, sample B was made of extracted cement of Nittetsu cement, and sample C was made of extracted cement of Nittetsu cement + AE water reducing agent.

The contribution by a single organic component to degree of deterioration (DD value) in the three samples (sample A, B, and C) can be calculated and was obtained by the following method using the DD values of the three types of mortar samples (samples A, B, and C).

In Table 1, the DD value of mortar sample A (Nittetsu cement) containing a very small amount of (phthalates + phosphate compounds) was 8%, and the DD value of sample B containing only (phosphate compounds) using the extracted cement was 4.9%, the DD value of mortar sample C containing (phosphate compounds + AE water reducing agent) made of the extracted cement with AE water reducing agent was 9.4%. The result is that the DD value of sample B (phosphate compounds) containing only a very small amount of phosphate compounds (DD value of phosphate: 4.9%), and the DD value with only a very small amount of AE water reducing agent in mortar sample C (phosphate compounds + AE water reducing agent) was 9.4%. Then, the DD value of only AE water reducing agent in sample C was 4.5%, obtained by subtracting 4.9% of the DD value of mortar sample B (phosphate compounds) from the 9.4% of the DD value of sample C (9.4%-4.9% = 4.5%).

The DD value (DD value: 8.0%) of the mortar A sample containing only a very small amounts of phthalates, reduced from the 8% of the DD value of mortar sample A (phthalates + phosphate compounds) the 4.9% of the DD value of mortar sample B (phosphate compounds) (8.0%-4.9% = 3.1%) and DD value of sample A became 3.1% and the 9.4% of the DD value of AE water reducing agent (sample C) was reduced as the DD value (4.9%) of sample B (9.4%-4.9% = 4.5%). Then, the DD value of sample C with only AE water reducing agent became 4.5%.

Further, the content ratio of phthalates in sample A to the cement was 0.0012% and if the content of phthalates (0.0012%) to the cement of Nittetsu cement were 1, the content of

organic substances in the phosphate compounds (0.12%) to cement corresponds to 100 times that of the phthalates and the content of AE water reducing agent (0.25%) corresponds to the cement as 208 times that of the phthalates. Now, the DD values for only one of three organic substances were the same order, namely 3.1% for the phthalates alone, 4.9% for the phosphate compounds alone, and 4.5% for the AE water reducing agent alone.

Referring to these results, it was concluded that even only one organic component in three types of organic matter (TQOM: phthalates, phosphate compound, and AE water reducing agent) small quantities of organic substance have a very significant influence on the deterioration of the mortar samples.

## 3.7 Pop-out of aggregate in concrete pavements

The pop-out of aggregate on the surface of a concrete airport runway was examined to elucidate the cause of the pop-out in this study. The concrete runway (A type concrete; runway: 2700 x 45 m) was constructed in Hokkaido, Japan in the fall of 2003. The following year (2004, 5–6 months after construction was completed), when snow melting agent was sprayed on the concrete pavement during the winter, a large amount of aggregate suddenly popped out from the surface of the pavement 4–5 hours after the spraying (pop-out of aggregate: 50,000 or more). At the time of the pop-out, parts of the mortar in the concrete runway changed to a very fine powder (disaggregation) and fine powder was also observed at the bottom of the pop-out aggregate holes (opened by the popped-out aggregate) in the concrete pavement.

The following experiments were conducted to elucidate this strange and superficially inexplicable phenomenon (pop-out of aggregate and disaggregation). The ethylene glycol-based snow melting agent sprayed here (1, 2-ethylene glycol: 88%, water: 9.5%) contained in a very small amount of organic substances such as two types of anionic surfactants (dibasic potassium phosphate; 1%; and sodium-di-(2-ethylhexyl) sulfosuccinate; 0.5%). From the results of the survey, there were mainly four kinds of pop-out aggregate: mud stone, tuffaceous siltstone, serpentine, and basalt-turned greenstone. A white fine substance (white fine powder of concrete: showing disaggregation) was everywhere observed in the original locations below the pop-out aggregate [4].

A part of the concrete pavement (A type concrete; thickness 30 cm) of the concrete runway was divided into small specimen (5 x 30 x 30 cm) and phenolphthalein solution applied to cross sections of this small specimen. As the result, the color of the entire cross section of the concrete runway remained colorless (pH <8). It shows that the concrete of the runway pavement had already deteriorated (deterioration). To elucidate the pop-out of aggregate on the concrete runway further, a small specimen was divided into 9 layers with the thickness of 2.5 cm. Two layers of the 9 layers (0–2.5 cm and 3–5.5 cm: sample: 2.5 x 2.5 x 8 cm) were collected to examine the cracks in these samples using a CT scanner (512 pixel).

The first layer (0–2.5cm) located at the surface of the concrete pavement and a second layer located at the depth of 3.0–5.5 cm were collected to examine the distribution of cracks in the small concrete sample in the vertical direction of the concrete runway, because the distribution of three-dimensional cracks in samples of the 3rd-9th layers showed similar crack distributions as that of second layer.

Fig 4 shows cracks in the aggregate and crack distribution by three-dimensional CT images of samples in the first and second layers.

Fig 4A and 4B show the cracks in the aggregate and cracks in the mortar in the first layer. Fig 4C and 4D show the cracks in the aggregate and cracks in the mortar in the second layer.

The aggregate types in the two layers (first layer and second layer) in Fig 4 is as follows.

GBS: Basalt-turned greenstone, GHY: Basalt-turned hyaloclastit,

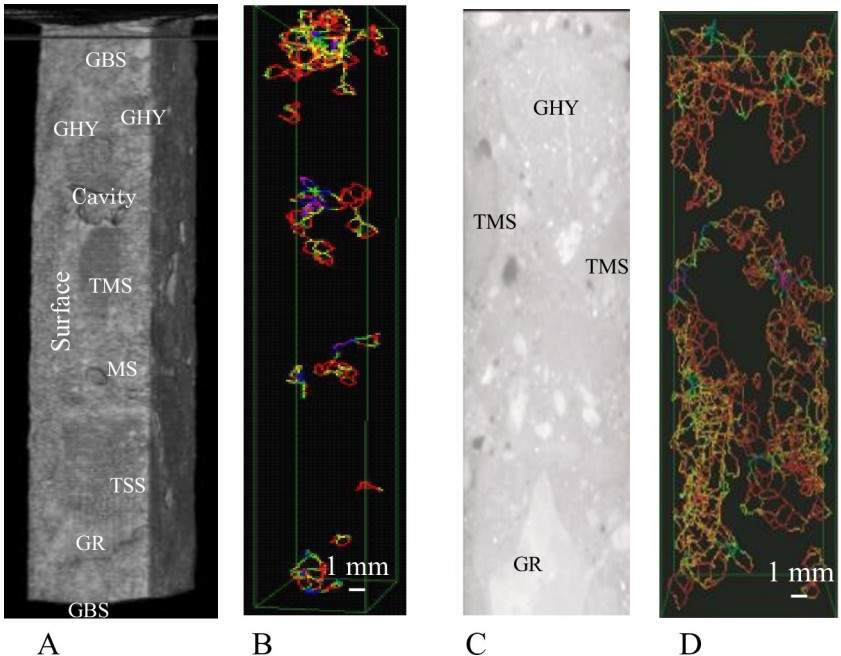

**Fig 4. Cracks in the damaged aggregate and crack distribution of three-dimensional cracks of the samples in the first (0–2.5 cm) and second layers (3–5.5 cm).**

TMS: Transformed mud stone, MS: Mud stone, TSS: Transformed sandstone, GR: Granite.

In Fig 4A–4C, the photos show CT images of the aggregate in the samples for the first and second layers. The color of the aggregate (TMS) in the sample in the first layer was somewhat blacker (darker) than that of the TMS in the second layer. It shows that the deterioration (amorphization) of aggregate of the sample in the first layer has proceeded further than that of the sample in the second layer.

All cracks in the first layer sample in Fig 4B concentrated inside and at the periphery of the aggregate, but no cracks were observed in the mortar part in sample of the first layer. Also, the inside of the aggregate GBS, TMS, and MS, of the sample in the first layer in Fig 4B has wider crack widths of 0.96 mm (blue), 1.2 mm (indigo) and 1.44 mm (purple). It shows that the aggregate has expanded. However, there were no cracks (Fig 4B) in the mortar in the first layer. Cracks with a narrow crack width (0.24 mm (red) and 0.72 mm (yellow)), spread throughout the sample and the maximum crack width (1.2 mm (indigo)) was rarely observed inside and in the periphery of the aggregate (TMS) here in the sample of the second layer. The crack distributions and crack widths of the samples of third to ninth layer in the samples were similar to that of the second layer shown in Fig 4D. It was assumed that the pop-out of aggregate was mainly caused by the volume expansion (cracks) of the aggregate in the first layer, because many cracks due to the surfactant for the snow melting agent concentrated especially inside and at the periphery of the aggregate (GBS, TMS, and MS). In addition, since cracks in the mortar part were few in the first layer (0–2.5 cm), it was supposed that a very small amount of organic matter (TQOM; phthalates, AE water reducing agent) had separated by vibration during sample preparation, and it is considered that this TQOM concentrated near the surface in the concrete runway. These results show an absence of cracks in the mortar parts of the sample in the first layer (0–2.5 cm) and that cracks of the aggregate occurred only in this concrete pavement in the runway. It suggests that this segregation (no crack in mortar in first layer) of organic matter (TQOM) is also related to the delamination (cracks) near the surface [4]. It

seems that the cracks in mortar parts of the sample obtained from the second to the ninth layer of samples were caused by organic matter of phthalates, phosphate compounds, and AE water reducing agent (TQOM) in the mortar, referring to segregation of organic matter due to vibration as described in section 3.4 (amorphous).

Further, the volume expansion of aggregate due to the surfactant in the snow melting agent was examined as follows.

The four types of aggregate where pop-out of aggregate had already occurred were immersed individually for a week in a 10% solution of each of two types of surfactants used in the snow melting agent sprayed on the concrete runway. As a result, a white gel-like substance generated on the surface of the aggregate with a thickness of 1–2 mm, and the color of the aqueous solution changed to slightly white smoky opaque. In the immersion test of the same kind of aggregate using a solution (10%) of 1, 2-ethylene glycol alone, there was no apparent change in the aggregate. For this reason, it may be assumed that the two types of anionic surfactants (dibasic potassium phosphate: 1%; and sodium-di-(2-ethylhexyl) sulfosuccinate: 0.5%) in the snow melting agent penetrated into the aggregate in the concrete runway in a very short time, and it seems that a part of the aggregate changed to an amorphous substance and caused the volume expansion, leading to the pop-out of the aggregate.

Further, when AE water reducing agent (Pozzolith No. 70) was added to concrete, the mortar part also expanded considerably as shown in Table 1 in Section 3.2 (Expansion), so the concrete had already expanded in the concrete pavement before the spraying with the snow melting agent. Since the harmful snow melting agent was sprayed with the concrete in this state, the two kinds of trace amounts of surfactants penetrated deep into the pavement, making the mortar and aggregate generated additional amorphous substance and caused further volume expansion of the pavement. It may be imagined that the compressive pressure due to volume expansion of the mortar and/or aggregate in concrete became greater. For this reason, it is thought that the un-expected pop-out of aggregate occurred due to an instantaneous large volume expansion of concrete.

The concrete pavement (A type concrete) on this runway was completely removed one year later due to the pop-out of the aggregate. The following year (2005), aggregate that did not pop out was selected, and all old concrete pavements (A Type concrete) were removed and replaced with new concrete pavements (Type B concrete). Here, there were only two types of aggregate in the new concrete runway, diabase (20–40 mm) and lime stone (5–20 mm), but the same AE water reducing agent, Nittetsu cement, and snow melting agent were also used for the new concrete runway (Type B concrete).

Subsequent observation of the new concrete pavement (Type B concrete) after spraying of the snow melting agent caused no pop-out of aggregate on the concrete runway with the two types of aggregate of this pavement (Type B concrete). However, the diabase aggregate disaggregated (granulated) on the surface of the pavement one year after the placement. However, with the limestone aggregate, no pop-out phenomenon and/or disaggregation phenomenon was observed on the surface of the concrete runway.

From the above results, the cause of the pop-out of aggregate generated on the concrete runway may be described as follows. Concrete had already partly deteriorated by small amounts of harmful organic matter (TQOM) in the cement just after placement, and amorphization of aggregate and volume expansion of mortar had occurred. For this reason, the snow melting agent containing surfactant with trace quantities of harmful organic matter was sprayed on the pavement surface where the mortar had expanded, and the amorphization of the mortar and aggregate progressed further, the volume expansion of the concrete runway progressed further. Therefore, the volume expansion and amorphization of the concrete runway (Type A concrete; runway) may be related to the pop-out of aggregate which has been considered to defy explanation so far.

## 3.8 Trace quantities of organic matter (TQOM)

Various samples were prepared in this study to identify organic matter in damaged concrete samples.

Many damaged concrete samples in France (N = 2), Belgium (N = 3) and Japan (N = 40) in 1997, including the damaged concrete samples (N = 30) obtained after the Great Hanshin Earthquake in Kobe in 1995 were collected and the $^1$H NMR test was performed for damaged concrete samples.

As a result, the peaks of phthalates (7.5 ppm, 7.6 ppm, 7.7 ppm), peaks of SPNES (3.5–4.4 ppm; oxyethylene moiety, -(O-CH$_2$CH$_2$)$_n$-) and peaks of asphalt of 0.9 ppm (CH$_3$), 1.2 ppm (CH$_2$), 1.5 ppm (CH) in the spectrum of $^1$H NMR for organic matters of all damaged concrete samples were detected and the waveforms of these spectra in the $^1$H NMR test showed very similar waveforms for all organic matter (phthalates, SPNES and asphalt) in the damaged samples (T. Tomoto, Canadian Journal of Civil Engineering, 35 (7) (2008) 744–750 [13]).

To further investigate for SPNES, windshield washer fluid, commercially available for automobiles was collected from around the world (United Kingdom, France, Belgium, Germany, Italia, Australia, Canada, USA, Japan, Taiwan, and Russia). The chemical composition of the surfactant in windshield washer fluid was investigated by $^1$H NMR Test. As a result, a chemical substance showing an oxyethylene moiety ((-O-CH$_2$CH$_2$)$_n$- of 3.5–4.2 ppm), like SPNES was detected in the organic matter of all windshields washer fluids.

This result suggests that SPNES and phthalates are present in the air worldwide. Based on this assumption, a comparative study was conducted on the TSM in the air (Sapporo, N = 1) and the organic matter of disaggregated concrete in the Kobe highway (N = 3). In this study, the organic matter of TQOM (TSM in Sapporo) in the air and organic matter in the disaggregated concrete samples in the Kobe highway were identified using HPLC, $^1$H NMR, and GC-MS. These chemical components for the samples of TSM, the disaggregated concrete in the Kobe highway, DEP, tire debris, and asphalt are shown in Table 3.

If asphalt pavements and concrete structures are breathing and they absorb trace quantities of organic matter (TQOM) in air, the chemical analysis of TQOM in the air may be very important as it can lead to damage to the structures where concrete was used.

**Table 3. Main chemical components, emission sources of the organic components of samples in disaggregated cement concrete, TSM, DEP, tire debris, SPNES and asphalt (bitumen).**

| Sample | Phthalates, Surfactant | R$_1$$^e$CONH$_2$ (amide) | Hydrocarbon |
|---|---|---|---|
| Disaggregated | DBP$^a$, DEHP$^b$, DOP$^c$ | C$_{15}$H$_{29}$, C$_{15}$H$_{31}$ | C$_{18}$H$_{38}$ |
| concrete | TMPDIB$^d$, SPNES$^f$ | C$_{17}$H$_{33}$, C$_{17}$H$_{35}$ | |
| TSM | DBP$^a$, DEHP$^b$, DOP$^c$ | C$_{15}$H$_{31}$, C$_{17}$H$_{33}$, C$_{17}$H$_{35}$ | C$_{18}$H$_{38}$ |
| | SPNES$^f$ | | |
| DEP | DBP$^a$, DEHP$^b$ | C$_{15}$H$_{31}$, C$_{17}$H$_{33}$, C$_{17}$H$_{35}$ | C$_{18}$H$_{38}$ |
| Tire | DBP$^a$, DEHP$^b$, DOP$^c$ | C$_{15}$H$_{31}$, C$_{17}$H$_{33}$, C$_{17}$H$_{35}$ | C$_{18}$H$_{38}$ |
| Asphalt | | C$_{17}$H$_{33}$, C$_{17}$H$_{35}$ | C$_{16}$H$_{34}$, C$_{17}$H$_{36}$ |
| (Bitumen) | | | C$_{22}$H$_{46}$, C$_{42}$H$_{86}$ |

(Source: Building and Environmnet,2009,**44**,2000–2005, Tomoto T. [5]).

a DBP: di-n-butyl phthalate.

b DEHP: di-(2-ethylhexyl) phthalate.

c DOP: di-octyl phthalate.

d TMPDIB: 2,2,4-trimethyl-1,3-pentanediol di-isobutylate.

e R$_1$: alkyl moiety.

f Sodium polyoxyethylene nonylphenyl ether sulfate (SPNES).

Multiple chemical substances in TQOM and TSM sample, DEP, tire debris, and asphalt were also identified to attempt to examine the emission sources of this organic matter using HPLC and GC-MS apparatus [14]. The emission sources and chemical components these samples are shown in Table 3.

Table 3 shows that TSM contained phthalates, amine compounds, and SPNES. These suggest that it is trace quantities of harmful organic matter in TQOM that cause damage to asphalt pavements and concrete structures (T. Tomoto, Building and Environment, 44 (2009) 2000–2005 [5]).

This study also describes individual emission sources of organic matter in TSM and disaggregated concrete, and these relationships between the organic matter of TSM and disaggregated concrete were examined using the chemical components in TSM and disaggregated concrete using Table 3. The same analysis (HPLC, GC-MS, $^1$H NMR) such as TSM and disaggregated concrete in the Kobe highway was also performed on the organic matter of DEP, asphalt, and tire debris. Phthalates were detected in all samples (disaggregated concrete in the Kobe highway, TSM, DEP, tire debris) except in the asphalt, and amine compounds were also detected in samples of disaggregated concrete in the Kobe highway, TSM, DEP, asphalt, and tire debris. From Table 3, the phthalates (DBP, DEHP, di-octyl phthalate (DOP)) in the chemical components in the disaggregated concrete from the Kobe highway included the same substances as the phthalates (DBP, DEHP, DOP) in the TSM sample.

From Table 3, four types of amine compounds of palmitoleamide ($C_{15}H_{29}CONH_2$), palmitamide ($C_{15}H_{31}CONH_2$), oleamide ($C_{17}H_{33}CONH_2$), and stearamide ($C_{17}H_{35}CONH_2$) were contained in the disaggregated concrete in the Kobe highway. The three types of amine compounds of palmitamide ($C_{15}H_{31}CONH_2$), oleamide ($C_{17}H_{35}CONH_2$) and stearamide ($C_{17}H_{35}CONH_2$) were also contained in the organic matter in the TSM sample in Sapporo. Table 3 shows that SPNES was included in TSM (Sapporo) and disaggregated concrete from the Kobe highway. The SPNES is the surfactant (0.5%) of windshield washer fluid of automobiles. It will be described in detail below. Peaks of chemical components of hydrocarbon ($C_{18}H_{38}$) in the $^1$H NMR test for disaggregated concrete of the Kobe highway and the substances of TSM were detected in samples of the disaggregated concrete from the Kobe highway and TSM.

Further, to estimate the emission sources of trace organic matter in the air (TSM), chemical components in DEP, tire debris, and asphalt were examined.

Table 3 suggests that the phthalates (DBP, DEHP, DOP) in the TSM in the air may be attributed to DEP or tire debris, and the di-octyl phthalate (DOP) of the phthalates in TSM may also be attributed to tire debris. However, palmitamide ($C_{15}H_{31}CONH_2$), oleamide ($C_{17}H_{33}CONH_2$) and stearamide ($C_{17}H_{35}CONH_2$) among the amine compounds in TSM are attributed to palmitamide ($C_{15}H_{31}CONH_2$), oleamide ($C_{17}H_{33}CONH_2$), and stearamide ($C_{17}H_{35}CONH_2$) in DEP and tire debris. However, only oleamide ($C_{17}H_{33}CONH_2$) and stearamide ($C_{17}H_{35}CONH_2$) among the amine compounds in asphalt were detected as amine compounds in TSM (Sapporo) in the air. Hydrocarbon ($C_{18}H_{38}$) in TSM (Sapporo) may be attributed to DEP or tire debris, but no hydrocarbon ($C_{18}H_{38}$) was found in the TSM (Sapporo) contained in the asphalt.

Table 3 shows emission source and chemical components in samples. However, the same organic phthalates (DBP, DEHP, and DOP), amine compounds (palmitamide, oleamide, and stearamide), and hydrocarbon ($C_{18}H_{38}$) were detected in the disaggregated concrete from the Kobe highway and TSM samples (Sapporo). This result suggests that trace quantities of organic matter (TQOM; phthalates, SPNES, amine compounds, and hydrocarbon) in the air are absorbed in concrete structures by the concrete respiratory action. It was reported in the results of the concrete slab damaged by small concentrations (1.6 g) of SPNES on the surface

of the asphalt pavement in the core sample of a new concrete bridge using repeated transient moisture permeation tests (46 times: it corresponds to 46 days) reported in section 3.11 [15]).

## 3.9 Respiratory action of concrete structures and asphalt pavements

Asphalt pavements and concrete structures have been reported to appear to take up atmospheric moisture together with trace quantities of harmful organic matter into their structures. Additionally, asphalt pavements and concrete structures seem to both absorb as well as adsorb atmospheric moisture and trace quantities of harmful organic matter during the respiratory action [16] and absorption and adsorption due to the respiratory action causes deterioration to the structures involved. Further, it has not been possible to distinguish and separate the absorption and adsorption in the respiratory process of asphalt pavements and concrete structures. For this reason, it has been unknown where in these structures the atmospheric moisture and trace quantities of harmful organic matter would accumulate when the structures deteriorate. The following transient moisture permeation test (summer condition) was conducted using glass wool insulation to resolve this problem. Glass wool insulation would not absorb the moisture as a sample of asphalt pavement or concrete structures. Glass wool insulation which is a commercially available high-density glass wool insulation (16 K/m$^3$) was used for this test.

Glass wool insulation, 10 cm thick, was packed into a cylindrical hollow container (diameter: 10 cm, length: 10 cm, wall thickness: 3 mm) made of vinyl chloride.

The transient moisture permeation tests were carried out in a summer condition with cylindrical containers covered by different films at the top and the bottom. The permeation conditions were simulated in summer condition by the dew condensation phenomenon in concrete bridges with water proofing sheets. The water content inside the glass wool insulation was measured by the weight (±0.1 g) of the glass wool insulation (total length: 10 cm) divided into three layers (length: 3 cm) using a microwave oven (500 W, 5 minutes).

Bottom film: poly ethylene (200 μm): moisture permeability resistance (m$^2$·S·Pa/ng): 0.410, moisture permeability (g/m$^2$·24h): 1.4. The top film: polyethylene with aluminum (9 μm), moisture permeability resistance (m$^2$·S·Pa/ng): 0.013, moisture permeability (g/m$^2$·24h): 45.0. As a result, the water content of upper part of the glass wool insulation polyethylene film with aluminum (9 μm) (outside buildings) was 0.034g (3.7%), the water content of center of the glass wool insulation (inside buildings) was 0.023g (2.5%), and the water content of lower part of the glass wool insulation with poly ethylene film (200 μm) (inside of building) was 0.868 g (93.8%). These films simulated with the skin sheet of glass wool insulation in house.

From this, it appears that glass wool insulation does not absorb moisture, but moisture is adsorbed in the glass wool insulation.

In the concrete slab of the concrete bridge (Tokachi bridge), the amount of organic matter was highest at the bottom (0.027%) and then near the surface (0.01%). A large peak showing SPNES in the spectrum of $^1$H NMR of concrete slab of Tokachi bridge was detected in the deep part of the slab rather than in the shallow part, indicating that SPNES penetrates deep into the concrete (T. Tomoto, Canadian Journal of Civil Engineering, 35 (7) (2008) 744–750 [13]).

The test results of the glass wool insulation using the transient moisture permeation test corresponds to the content of organic matter with water due to the respiration of concrete structures stored near the surface and bottom of concrete slabs. These results suggest that organic matter in concrete depends both on absorption as well as on adsorption.

The results show that much moisture was adsorbed at the bottom as well as during the passage of the moisture and that moisture easily penetrated to the polyethylene film processed with aluminum. Sasaki performed transient moisture permeation test under summer

condition for the core sample of new concrete taken from bridge. He found many drips of water stored through of transparent vinyl film in the bottom of the slab of core samples (diameter 10 cm, thickness: 13 cm) of a newly constructed concrete bridge with a proofing layer and two layered-asphalt mixtures (thickness: 7 cm) (I. Sasaki, Journal of the Japan Petroleum Institute, 49 (6) (2006) 315–320 [15]).

These results suggest that a small amount of moisture together with trace quantities of harmful organic matter in the air penetrated the water proofing sheet in the concrete bridge by respiratory action (absorption and adsorption) and moisture accumulated at the bottom of the concrete slab in the summer condition, referring to the experiment with glass wool insulation. Trace quantities of harmful organic matter in TSM contained included SPNES and phthalates contained in the air. It is possible that harmful moisture such as SPNES and phthalates in the air caused the damage such as disaggregation during respiratory action of asphalt pavements and concrete structures.

## 3.10 Organic matter in damaged concrete structures and damaged asphalt pavements

Asphalt pavements and concrete structures appear to show respiratory action, as described in section 3.9. If these structures were shown to be subject to respiratory action, it was presumed that moisture and TQOM in the air were taken up in the structures. Sasaki conducted transient moisture permeation tests in the summer condition using damaged core samples (These damaged samples are described next. diameter: 10 cm) of asphalt pavements. Sasaki performed transient moisture permeation tests for core samples of blistered and rutted areas (severe flow of surface course of asphalt pavement) which had occurred on the runway of the asphalt pavement in Nagoya and for core samples of highways in service. From this, it was confirmed that the moisture in the air was taken into the core samples as water. It seems that the blistering at the interface of the surface course and base course in the asphalt pavement were caused by harmful water [17]. It shows that the water was also stored at the interface between the surface and base courses in asphalt pavements, because asphalt emulsion (cation type) of the interlayer between the surface and base courses in asphalt pavements at blistered areas was completely dissolved by SPNES (anion type) which penetrated from in the air and moisture stored as water in the interlayer in the asphalt pavement.

Disaggregation of the asphalt runway in Nagoya related to SPNES in the asphalt runway and it will be described in section 3.11.

Peel-off of the surface course in the asphalt runway in Nagoya was caused by separation between the surface course and base course in the asphalt pavements and shearing forces due to airplanes. The water in the interlayer changed gas at the high temperatures in summer and it easily evaporate. When the gas at the interface between surface course and base course swells by the high temperatures in summer, blistering was caused by the gas pressure.

Sasaki also collected a core samples (diameter: 10 cm) from a new concrete bridge with two layered-asphalt pavement and water proofing sheet, placed a very small amount of SPNES (1.6 g) on the surface of the asphalt pavement of this new core sample, and conducted repeated (46 times) transient moisture permeation tests (the repeated transient moisture permeation test; 8-hour cycle test in the following) under the summer condition. As a result, it was confirmed that the water content of the core sample increased linearly with increases and decreases (inhaled and exhaled: termed respiratory action) with the number of repetitions of the test (water content at repeated test of 46 times: total 20 g), and that the SPNES penetrated the water proof sheet, and the concrete slab deteriorated by penetration of SPNES, and bitumen which has peaks of 0.9 ppm ($CH_3$), 1.2 ppm ($CH_2$), and 1.5 ppm (CH) was also detected in the

spectrum of ${}^1$H NMR Test for organic matters of the powdered sample of the core sample of this damaged concrete slab (thickness: 3.5 cm) after the transient moisture permeation test. It shows that SPNES in the air affected the damage to the asphalt pavement and concrete slab in the concrete bridge [3].

Therefore, the inexplicable black spots seen at the cracks on the back of disaggregated concrete slabs in concrete bridges in the field may be thought to be a phenomenon where the asphalt pavement was dissolved with SPNES in the air and it was exuded outside of the concrete slab.

From the above experimental results, it was suggested that harmful water stored in asphalt pavements and concrete structures together with trace quantities of harmful organic matter in the air inside of these structures by respiratory action in the summer condition. The content of organic material in the concrete structures could easily be measured using the ratio of organic matter to mortar for powdered samples of damaged concrete. Also, the water content in the surface course and/or base courses in damaged asphalt pavements could easily be measured instead of trace of harmful organic in asphalt pavements.

The water content in asphalt mixtures stored for two years was measured to investigate the damage at a runway in Nagoya in 2000. The asphalt-paved runway in Nagoya was provided with a top course twice (construction of two layered-asphalt overly: 1960, 1998). After the second top course placement, the disaggregation in the base course of the asphalt pavement occurred in the runway (2000). Asphalt pavements are formed by asphalt mixtures made at high temperatures, 150˚C, which result in the water content just after its placement becoming less than 0.5%. As a result, water detected at concentrations higher than 0.5% of water in the asphalt mixtures may be assumed to have been absorbed moisture from in the air.

The water content in the surface course and base course after placement of two layered-overly at 1998 of asphalt runway of Nagoya was investigated in 2000.

Table 4 shows that the water contents in the base course (3.3%, first overly was added:1960; 2.7%, second overly was added: 1998) of the asphalt runway pavement were higher than those of the original surface course (1%, construction of overlay: 1960, 2%, construction of overlay: 1998), suggesting that the moisture from the air stored inside the bottom (base course) of the asphalt pavement.

Next, the following damaged core samples in asphalt pavement (two-layered system) were used to investigate the relationship between residual water content and damage in the samples after the transient moisture permeability test. In 2000, four types of core samples (diameter: 10 cm) were also collected from the three types of damaged areas in the runway pavement (areas with blistering, rutting, and runway in service) in Nagoya to examine the respiratory action of the surface course and binder course added in 1998, where blistering and rutting were remarkable, runway in service and a highway (one type) in service area in 2000, and a transient moisture permeation test with four types of core samples of asphalt pavements in Table 4 was performed under the summer condition and residual water was measured. The transient moisture permeation test for the four samples were performed to identify unexplained disaggregation and blistering of asphalt pavement in the runway in Nagoya.

Table 4 shows that the amount of these residual water contents (water content) after the transient moisture permeation test for core samples (surface course and binder course were added: 1998) of three areas of the asphalt runway (water content: blistering; 6.5%, rutting; 4.7%, runway in service; 15.5%) was larger than the water content (0.6%) in the core sample of the highway.

This result in Table 4 suggests that the larger water content of the asphalt pavement in the disaggregated runway in Nagoya affects the disaggregation. This disaggregation of asphalt pavements in the runway of Nagoya will be described further in section 3.11.

Table 4. Properties of asphalt mixtures in runway in Nagoya.

| Layer | airport runway (Nagoya) | | | | highway (Hokkaido) | |
|---|---|---|---|---|---|---|
| | surface course** | base course** | surface course* | base course* | surface course | base course |
| | construction: 1960 | | construction: 1998 | | | |
| Thickness of layer (cm) | 4 | 17 | 3 | 9 | 5 | 7 |
| Type of mixture | dense | coarse | dense | coarse | dense | coarse |
| Max size of aggregate (mm) | N/A | N/A | 20 | 20 | 13 | 20 |
| Density (g/cm$^3$) | N/A | N/A | 2.39 | 2.35 | 2.35 | 2.4 |
| Air void (%) | N/A | N/A | 3 | 5.4 | 4.7 | 4.7 |
| Binder content (%) | N/A | N/A | 5.4 | 4.6 | 5.5 | 4 |
| Asphalt binder | N/A | N/A | StAs60/80 | StAs60/80 | StAs80/100 | StAs80/100 |
| Water content of mixture (%) | 1** | 3.3** | 2* | 2.7* | - | - |
| Original water content of mixture (%) | <0.5 | <0.5 | <0.5 | <0.5 | <0.5 | <0.5 |
| Residual water content after moisture permeation test (g) (24 hours at summer condition) | | | core sample: (3+9) cm* | | core sample: (5+7) cm | |
| | | | blisterling | 6.5 | | |
| | | | rutting | 4.7 | | |
| | | | runway | 15.5 | | 0.6*** |

*: Construction: 1998 (Investigation: 2000).

**: Construction: 1960 (Investigation: 2000).

***: Highway in service.

In order to investigate the distribution of organic matter in the depth direction of the concrete structure, the concrete slab (thickness: 30 cm) in the Tokachi bridge (constructed in 1934) was also examined. This concrete slab (30 cm) was divided into eight layers each with a thickness of 3 cm. Split tensile test was performed for each layer and after the test these samples were crushed under 0.074 mm or smaller. These powdered samples were used to measure the amount of organic matter to the mortar of each layer was investigated to examine the damage to the concrete using the powdered-samples of thickness of 3 cm.

Fig 5 shows the ratio of the content of organic matter to mortar for the concrete slab, foundation of the handrail, and bridge supporting pier.

No.1 is for the upper part of the foundations of the hand rail, No. 2 is for the bottom part of the foundations of the hand rail. The samples of the bridge pier were collected from the same bridge pier, and the air means the part of the bridge pier (air) exposed to the air, and the water means the part of the bridge pier (water) that was submerged in the river.

Fig 5 shows the content of organic matter accumulated in the bottom in the concrete slab. In this concrete structure (concrete slab, foundations of handrail, and bridge supporting pier) of the same concrete bridge (Tokachi bridge), they showed different ratios of organic matter to mortar. The chemical peaks of phthalates and SPNES in the spectrum of $^1$H NMR Test were detected in these samples.

From the spectrum of the $^1$H NMR Test for this slab, SPNES penetrates deeper into the concrete slab than phthalates. This suggests that SPNES is a surfactant that easily penetrates even in asphalt pavements, so it is likely that it penetrates deeply (T. Tomoto, Canadian Journal of Civil Engineering, 35 (7) (2008) 744–750 [13]).

The results suggested that the organic matter and/or water accumulated in the bottom parts of these asphalt pavements and concrete structures.

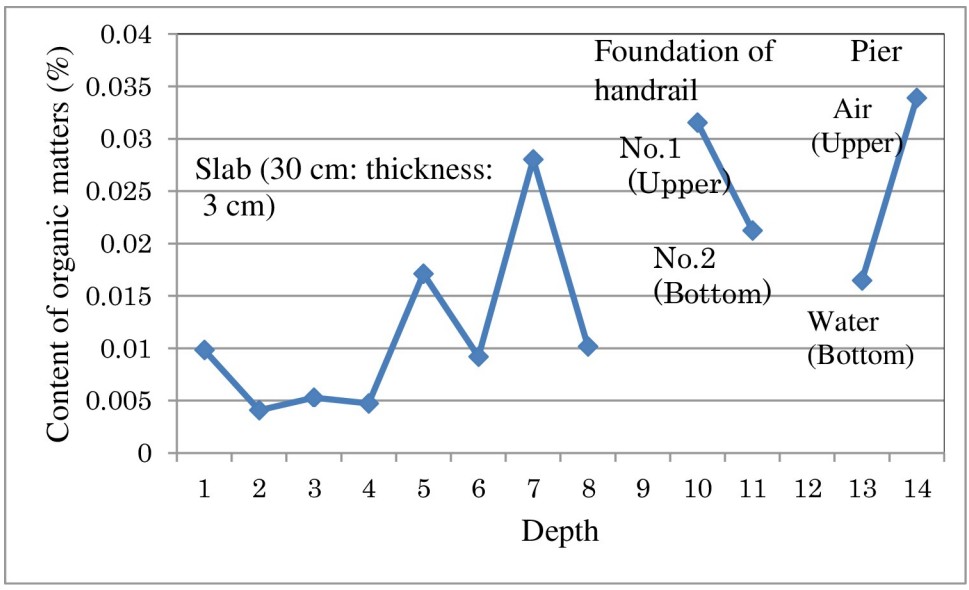

**Fig 5. Distribution of organic matter contents of concrete slab, foundations of handrail, and bridge supporting pier.**

### 3.11 Disaggregation in asphalt pavements and concrete structures

We found that disaggregation had occurred in these structures (disaggregated concrete in Kobe highway and asphalt runway of Nagoya).

For this reason, the causes of the disaggregation in the concrete structures and asphalt pavements were examined for disaggregated concrete in the Kobe highway and the disaggregated asphalt runway in Nagoya in this study.

Some parts of the disaggregated concrete in the Kobe highway had changed to fine powder (0.074 mm or smaller) in 1994. These structures were made 30 years ago. It shows that disaggregation of concrete had occurred.

Air pollution in the Kobe highway area was particularly severe in Japan in 1995 and there were many patients with respiratory diseases in this area of Kobe (A. Moriyoshi, Journal of the Japan Petroleum Institute, 45 (2) (2002) 84–88 [3]). Kobe city is located on a narrow area where two main roads running east and west are sandwiched between north, mountains, and south, the sea. There were two main roads along the city: a highway (160,000 vehicles/day, 10 lanes) and an elevated highway of the Kobe highway (40,000 vehicles/day, 4 lanes), and these roads were always crowded in 1994 (A. Moriyoshi, Journal of the Japan Petroleum Institute, 45 (2) (2002) 84–88 [3]).

Concrete samples (N = 3) of the disaggregated concrete in the Kobe highway were collected in 1995 and these concrete samples turned into fine powder (less than 0.074 mm) using crushed with a hammer. The method of the chemical analysis and chemical components in these structures are described in section 2.2.3.

Organic substances were extracted from the powdered-samples using Soxhlet extractor and benzene-methanol solution, and HPLC, GC-MS, and $^1$H NMR tests were performed on the samples to identify the chemical components of organic substances in the samples.

The following organic substances were identified and they are shown in disaggregated concrete in Table 3 in section 3.8 (Table 3). From these polar components, phthalates (DBP, DEHP, and di-octyl phthalate (DOP)), four kinds of amine compounds (palmitoleamide,

palmitamide, oleamide and stearamide), TMPDIB, SPNES and hydrocarbons were detected and they are detailed in Table 3. These organic matters are harmful organic matter in concrete structures. The ratio of organic matter to mortar in the disaggregated concrete in the Kobe highway was 0.2%. It was suggested that the disaggregated concrete structures in the Kobe highway contained TQOM (phthalates, SPNES, amine compounds), which was taken from the air and they played a role in the disaggregation of the concrete structures.

The asphalt pavement runway in Nagoya was constructed in 1998, and two years later, much blistering, peel-off of surface course and disaggregation (depth: 15 cm) in the runway of the asphalt pavement were observed in the summer of 2000. At the areas where blistering occurred at the interface between the surface course and base courses, the tack coat (black membrane) using asphalt emulsion, which adhered to the surface course and base courses, had completely dissolved and disappeared. This phenomenon indicates that harmful water accumulated at the interface of the surface course and base course of the asphalt pavement and harmful water dissolved the tack coat. It was considered that SPNES in the air penetrated by respiratory action of structures to inside of structures and SPNES in the air dissolved the tack coat, because SPNES is the surfactant of anion type and tack coat is the cation type of asphalt emulsion. Then, it seems that peel-off of the surface course in the runway was caused by dissolution of the tack coat and shearing forces of airplanes.

The following experiment was performed to confirm the existence of SPNES in the disaggregated asphalt pavement runway in the Nagoya.

A core sample with a diameter of 10 cm (length: 15 cm) was taken from this damaged runway in Nagoya in 2000 (two layered-asphalt overly: 1998) to examine the organic matter (SPNES and phthalates) in the asphalt pavement, and the core sample was divided into the five of 2.5 cm thick layers. Organic matter in the samples of the five layers was extracted using a Soxhlet extractor and chloroform solution. (A. Moriyoshi, Journal of the Japan Petroleum Institute, 45(12) (2002), 84–88 [3]).

The spectrum of $^1$H NMR for organic matter of the five layers had very similar waveforms and they showed the same peaks of organic matter. The peak of SPNES in spectrum of $^1$H NMR Test (oxyethylene moiety: $(-O-CH_2CH_2)_n-$: 3.5–4.2 ppm), and the peaks of bitumen at 0.9 ppm ($CH_3$), 1.2 ppm ($CH_2$) and 1.5 ppm (CH) and the peaks of the phthalates at 7.5 ppm, 7.7 ppm, and 4.3 ppm were also detected in the samples of all five layers. It suggests that disaggregation in asphalt runway in Nagoya was caused by SPNES.

As phthalates and SPNES were also detected in all samples of the five layers of disaggregated asphalt pavement of the Nagoya runway and disaggregated concrete structures in Kobe highway, it was concluded that the disaggregation of disaggregated asphalt pavement runway in the runway in Nagoya and the disaggregated concrete structure in Kobe highway were related to disaggregation due to TQOM in the air.

The causes of the accumulation of TQOM and moisture inside disaggregated concrete structures and disaggregated asphalt pavement are as follows.

When the air temperature and the relative humidity in the air changes with time, the concrete structures and the asphalt pavements repeatedly inhaled and exhaled (respiratory action: once a day), and harmful organic matter in the air accumulated together with the number of respiratory actions in the summer condition. It was considered that these harmful traces of organic matter and moisture in the air caused the various kinds of damage (disaggregation, deterioration, blistering, peel-off of surface course, amorphization and crack) to the concrete structures and asphalt pavements.

These so far unexplained damage deterioration can be expected to be particularly severe in concrete structures and asphalt pavements in areas where the air is severely polluted by traveling vehicles.

## 4. Conclusions

Following conclusions were obtained in this study.

1. All causes of various strange damages in concrete structures and asphalt pavements are trace amounts of organic matter. They damage to these structures in a very short period of time.

2. Trace of quantities of organic substances in cement existed. They are phthalates compounds phosphate compounds and AE water reducing agent. These are derived from waste tires, waste plastics and meat-and-bone meal in process of manufacture of cement.

3. Small amounts of tire dust, DEP, snow melting agents, SPNES, and asphalt are suspended in the air. Since these organic substances are taken into the inside by the respiratory action of these structures, these structures were damaged by these organic substances.

4. The crack width and crack length are the decisive factors for concrete damage rather than deterioration.

5. DD Value using CT is effective index for evaluating the deterioration of concrete.

6. The damage of concrete structures and asphalt pavements related with the amount of organic matter inside the former and the amount of water inside the latter, respectively.

## Supporting information

**S1 File.**
(DOCX)

## Acknowledgments

The authors wish to express gratitude to Professor M. Tabata, Dr. T. Tomoto, Mr. K. Taki, Dr. H. Ishikawa, Dr. K. Tokumitsu, Professor H. Takahashi, Dr. Y. Kusashima, Dr. S. Ono, Mr. Y. Chinzei, Mr. M. Uesaka, and to Hokkaido Regional Development Bureau for providing valuable 120 year-sample and to Hokkaido Research Organization for utilization of CT scanner. Professor M. Kamijima in Nagoya City University gave helpful suggestions to the authors.

## Author Contributions

**Conceptualization:** Akihiro Moriyoshi.

**Formal analysis:** Akihiro Moriyoshi, Masahito Natsuhara, Kiyoshi Sakai, Takashi Kondo, Akihiko Kasahara.

**Supervision:** Eiji Shibata.

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
