## [Decision Letter · Decision Letter 0]

18 Jan 2021

PONE-D-20-22508

Modern cement concrete and asphalt pavement are damaged by respiratory action and a trace quantities of organic matter (TQOM)

PLOS ONE

Dear Dr. moriyoshi,

Thank you for submitting your manuscript to PLOS ONE. After careful consideration, we feel that it has merit but does not fully meet PLOS ONE’s publication criteria as it currently stands. Therefore, we invite you to submit a revised version of the manuscript that addresses the points raised during the review process.

Although the study is new and important. However, as a matter of fact, the paper is unnecessarily lengthy and can be shortened by providing the names of the test methods with proper references. A list of abbreviations can also be provided for the interest of the readers. Also, eliminate the lengthy test procedures details in results and discussion part. 

We look forward to receiving your revised manuscript.

Kind regards,

Anwar Khitab

Academic Editor

PLOS ONE

Journal Requirements:

2. Please ensure that in your methods section you have provided details of the sources of all materials, chemicals, equipment and instrumentation used in your study, including manufacturer names where relevant. This is in line with our reproducibility criterion for publishing, see https://journals.plos.org/plosone/s/criteria-for-publication#loc-3

3.We note that you have indicated that data from this study are available upon request. PLOS only allows data to be available upon request if there are legal or ethical restrictions on sharing data publicly. For more information on unacceptable data access restrictions, please see http://journals.plos.org/plosone/s/data-availability#loc-unacceptable-data-access-restrictions.

4.Thank you for stating the following in the Financial Disclosure section:

"The authors recieved no specific funding for this work."

We note that one or more of the authors are employed by a commercial company: Green Consultant Co. Ltd. and Shimadzu Corporation

Additional Editor Comments:

In general, the manuscript contains significant numbers of variables: Therefore, it is advised to present a list of abbreviations for the interest of the readers.

Also, it is advised to shorten the manuscript by elaborating the name of the test method with proper references. Explaining the whole test from samples to test conditions, the brand names of the apparatus again and again has made this important study unnecessary lengthy.

Abstract:

Abstract should be concise, with brief problem statement, methodology, results and conclusion. In the current abstract, most of the part is introduction of the problem. It is suggested to exclude large introduction part and concentrate on brief methodology, results and conclusions as per standard format of an abstract.

Introduction:

Line 97: Write the complete name once for “AE”, where it appears for the first time.

In general, the introduction part lacks some important literature review related to the study.

Experiments and test methods:

Line 198: Change the word “Europa” with “Europe”. (I think so).

Line 199: Kindly write full name for “TSM”, “SPNES”, when they appear for the first time OR separately "Provide a list of abbreviations" and then do not use the whole names throughout the manuscript. At the moment, it is mix-up.

Line 218: Kindly Write fully explained term for “1H NMR”. This will be otherwise confusing for the readers. You may alternatively write “Nuclear Magnetic Resonance test”. Similar complete names can be written for 2.2.2, and 2.2.3.

Line 233: What is meant by “(Themes: 1, 2, 3, 4, 5, 6, 7, 8, 9, 10 ,11) “? Kindly omit all such themes throughout the manuscript as they are not understandable at all.

As a whole, the test procedures are described in detail. In my humble view, brief description with some reference is sufficient for the interest of the readers.

Results and Discussion:

It is better to indicate the test methods with reference rather than the brand names of the instrument.

Heading 3.1: Can the observations be presented in tabular form rather than paragraphs?

Line 391: Kindly elaborate “2E1H” for the first time.

Line 395: What is meant by “This gas”?

Line 412: Change “investigate” with “investigated”.

Kindly elaborate “DD” as “Degree of Deterioration”, when it appears for the first time.

Kindly enhance font size in Table 1.

Test methods are repeatedly presented: It is suggested to present the names of the tests and their references rather than explaining the tests again and again. This has made the manuscript very lengthy.

Conclusions:

Conclusions are somewhat dry. Line 1275 to 1286 can be added to the conclusions rather than being the part of the results and discussion.

Reviewers' comments:

Reviewer's Responses to Questions

**Comments to the Author**

1. Is the manuscript technically sound, and do the data support the conclusions?

Reviewer #1: Yes

2. Has the statistical analysis been performed appropriately and rigorously? 

Reviewer #1: I Don't Know

3. Have the authors made all data underlying the findings in their manuscript fully available?

Reviewer #1: Yes

4. Is the manuscript presented in an intelligible fashion and written in standard English?

Reviewer #1: No

5. Review Comments to the Author

Reviewer #1: Paper is very lengthy and readers won’t be able to understand the clear theme of the paper as contents are irregularly arranged.

The authors didn’t mention any details regarding 1H NMR, GC-MS (JMS-AX-500), HPLC, High Volume Sampler, One dimensional Transient moisture permeation test and Micro focus CT scanning (CT) in the Introduction section. It’s hard for the reader to pick new terms later in the paper.

I believe that this paper would be of interest to highway engineers. Therefore, make it simple to read.

6. PLOS authors have the option to publish the peer review history of their article (what does this mean?). If published, this will include your full peer review and any attached files.

Reviewer #1: No

---

## [Author Response · Author response to Decision Letter 0]

13 Feb 2021

I appreciate with many reviewers.

---

## [Editor Report · Decision Letter 1]

16 Mar 2021

PONE-D-20-22508R1

Modern cement concrete and asphalt pavement are damaged by respiratory action and trace quantities of organic matter (TQOM): Civil structures are damaged by respiratory action and trace quantities of organic matter

PLOS ONE

Dear Dr. moriyoshi,

Thank you for submitting your manuscript to PLOS ONE. After careful consideration, we feel that it has merit but does not fully meet PLOS ONE’s publication criteria as it currently stands. Therefore, we invite you to submit a revised version of the manuscript that addresses the points raised during the review process.

Kindly change the title. Presently, it is in sentence format. Change it to the title format. May I suggest the following title.

"Deterioration of modern concrete and asphalt pavements by respiratory action and trace quantities of organic matter". I also think there is no need for a short title, so that can be omitted.

We look forward to receiving your revised manuscript.

Kind regards,

Anwar Khitab

Academic Editor

PLOS ONE

Journal Requirements:

Additional Editor Comments (if provided):

Kindly change the title. Presently, it is in sentence format. Change it to the title format. May I suggest the following title.

"Deterioration of modern concrete and asphalt pavements by respiratory action and trace quantities of organic matter". I also think there is no need for a short title, so that can be omitted.

---

## [Author Response · Author response to Decision Letter 1]

23 Mar 2021

I am very grateful to Dr. Preston for his encouragement and suggestions for the various new discoveries and the structure of the new treatise. We would also like to thank Dr. Khitab and Dr. Magyar for their long-standing advice, proofreading, and support for this treatise.

I think this treatise has become very long and difficult to peer review. We would like to thank all the people concerned.

---

## [Editor Report · Decision Letter 2]

25 Mar 2021

Deterioration of modern concrete structures and asphalt pavements by respiratory action and trace quantities of organic matter

PONE-D-20-22508R2

Dear Dr. moriyoshi,

We’re pleased to inform you that your manuscript has been judged scientifically suitable for publication and will be formally accepted for publication once it meets all outstanding technical requirements.

Kind regards,

Anwar Khitab

Academic Editor

PLOS ONE

Additional Editor Comments (optional):

Thanks for the understanding and patience.
---

## [Editor Report · Acceptance letter]

19 Apr 2021

PONE-D-20-22508R2 

Deterioration of modern concrete structures and asphalt pavements by respiratory action and trace quantities of organic matter

Dear Dr. Moriyoshi:

I'm pleased to inform you that your manuscript has been deemed suitable for publication in PLOS ONE. Congratulations! Your manuscript is now with our production department. 

Kind regards, 

on behalf of

Dr. Anwar Khitab 

Academic Editor

PLOS ONE